# Self-Supervised GANs with Label Augmentation

**Liang Hou**[1,3]**, Huawei Shen**[1,3]**, Qi Cao**[1]**, Xueqi Cheng**[2,3]

[1]Data Intelligence System Research Center,
Institute of Computing Technology, Chinese Academy of Sciences
[2]CAS Key Laboratory of Network Data Science and Technology,
Institute of Computing Technology, Chinese Academy of Sciences
[3]University of Chinese Academy of Sciences
{houliang17z,shenhuawei,caoqi,cxq}@ict.ac.cn

## Abstract

Recently, transformation-based self-supervised learning has been applied to generative adversarial networks (GANs) to mitigate catastrophic forgetting in the discriminator by introducing a stationary learning environment. However, the separate self-supervised tasks in existing self-supervised GANs cause a goal inconsistent with generative modeling due to the fact that their self-supervised classifiers are agnostic to the generator distribution. To address this problem, we propose a novel self-supervised GAN that unifies the GAN task with the self-supervised task by augmenting the GAN labels (real or fake) via self-supervision of data transformation. Specifically, the original discriminator and self-supervised classifier are unified into a label-augmented discriminator that predicts the augmented labels to be aware of both the generator distribution and the data distribution under every transformation, and then provide the discrepancy between them to optimize the generator. Theoretically, we prove that the optimal generator could converge to replicate the real data distribution. Empirically, we show that the proposed method significantly outperforms previous self-supervised and data augmentation GANs on both generative modeling and representation learning across benchmark datasets.

## 1 Introduction

Generative adversarial networks (GANs) [14] have developed rapidly in synthesizing realistic images in recent years [4, 23, 24]. Conventional GANs consist of a generator and a discriminator, which are trained adversarially in a minimax game. The discriminator attempts to distinguish the real data from the fake ones that are synthesized by the generator, while the generator aims to confuse the discriminator to reproduce the real data distribution. However, the continuous evolution of the generator results in a non-stationary learning environment for the discriminator. In such a non-stationary environment, the single discriminative signal (real or fake) provides unstable and limited information to the discriminator [37, 7, 70, 57]. Therefore, the discriminator may encounter catastrophic forgetting [25, 6, 55], leading to the training instability of GANs and thereby mode collapse [52, 61, 33, 36] of the generator in learning high-dimensional, complex data distributions.

To alleviate the forgetting problem of the discriminator, transformation-based self-supervised tasks such as rotation recognition [13] have been applied to GANs [6]. Self-supervised tasks are designed to predict the constructed pseudo-labels of data, establishing a stationary learning environment as the distribution of training data for this task does not change during training. The discriminator enhanced by the stationary self-supervision is able to learn stable representations to resist the forgetting problem, and thus can provide continuously informative gradients to obtain a well-performed generator. After obtaining empirical success, self-supervised GAN was theoretically analyzed and partially improved by a new self-supervised task [57]. However, these self-supervised tasks cause a goal inconsistent with

35th Conference on Neural Information Processing Systems (NeurIPS 2021).

generative modeling that aims to learn the real data distribution. In addition, the hyper-parameters that trade-off between the original GAN task and the additional self-supervised task bring extra workload for tuning on different datasets. To sum up, we still lack a proper approach to overcome catastrophic forgetting and improve the training stability of GANs via self-supervised learning.

In this paper, we first point out that the reason for the presence of undesired goal in existing self-supervised GANs is that their self-supervised classifiers that train the generator are unfortunately agnostic to the generator distribution. In other words, their classifiers cannot provide the discrepancy between the real and generated data distributions to teach the generator to approximate the real data distribution. This issue originates from that these classifiers are trained to only recognize the pseudo-label of the transformed real data but not the transformed generated data. Inspired by this observation, we realize that the self-supervised classifier should be trained to discriminatively recognize the corresponding pseudo-labels of the transformed generated data and the transformed real data. In this case, the classifier includes the task of the original discriminator, thus the original discriminator could even be omitted. Specifically, we augment the GAN labels (real or fake) via self-supervision of data transformation to construct augmented labels for a novel discriminator, which attempts to recognize the performed transformation of transformed real and fake data while simultaneously distinguishing between real and fake. By construction, this formulates a single multi-class classification task for the label-augmented discriminator, which is different from a single binary classification task in standard GANs or two separate classification tasks in existing self-supervised GANs. As a result, our method does not require any hyper-parameter to trade-off between the original GAN objective and the additional self-supervised objective, unlike existing self-supervised GANs. Moreover, we prove that the optimal label-augmented discriminator is aware of both the generator distribution and the data distribution under different transformations, and thus can provide the discrepancy between them to optimize the generator. Theoretically, the generator can replicate the real data distribution at the Nash equilibrium under mild assumptions that can be easily satisfied, successfully solving the convergence problem of previous self-supervised GANs.

The proposed method seamlessly combines two unsupervised learning paradigms, generative adversarial networks and self-supervised learning, to achieve stable generative modeling of the generator and superior representation learning of the discriminator by taking advantage of each other. In the case of generative modeling, the self-supervised subtask resists the discriminator catastrophic forgetting, so that the discriminator can provide continuously informative feedback to the generator to learn the real data distribution. In the case of representation learning, the augmented-label prediction task encourages the discriminator to simultaneously capture both semantic information brought by the real/fake-distinguished signals and the self-supervised signals to learn comprehensive data representations. Experimental results on various datasets demonstrate the superiority of the proposed method compared to competitive methods on both generative modeling and representation learning.

## 2 Preliminaries

### 2.1 Generative Adversarial Networks

Learning the underlying data distribution $\mathcal{P}_d$, which is hard to specify but easy to sample from, lies at the heart of generative modeling. In the branch of deep latent variable generative models, the generator $G : \mathcal{Z} \to \mathcal{X}$ maps a latent code $z \in \mathcal{Z}$ endowed with a tractable prior $\mathcal{P}_z$ (e.g., $\mathcal{N}(0, \mathrm{I})$) to a data point $x \in \mathcal{X}$, and thereby induces a generator distribution $\mathcal{P}_g = G \# \mathcal{P}_z$. To learn the data distribution through the generator (i.e., $\mathcal{P}_g = \mathcal{P}_d$), one can minimize the Jensen–Shannon (JS) divergence between the data distribution and the generator distribution (i.e., $\min_G \mathbb{D}_{\mathrm{JS}}(\mathcal{P}_d \| \mathcal{P}_g)$). To estimate this divergence, generative adversarial networks (GANs) [14] leverage a discriminator $D : \mathcal{X} \to \{0, 1\}$ that distinguishes the real data from the generated ones. Formally, the objective function for the discriminator and generator of the standard minimax GAN [14] is defined as follows:

$$\min_G \max_D V(G, D) = \mathbb{E}_{x \sim \mathcal{P}_d}[\log D(1|x)] + \mathbb{E}_{x \sim \mathcal{P}_g}[\log D(0|x)], \qquad (1)$$

where $D(1|x) = 1 - D(0|x) \in [0, 1]$ indicates the probability that a data $x$ is distinguished as real by the discriminator. While theoretical guarantees hold in standard GANs, the non-stationary learning environment caused by the evolution of the generator contributes to catastrophic forgetting of the discriminator [6], which means that the discriminator forgets the previously learned knowledge by over-fitting the current data and may therefore hurt the generation performance of the generator.

## 2.2 Self-Supervised GAN

To mitigate catastrophic forgetting of the discriminator, self-supervised GAN (SSGAN) [6] introduced an auxiliary self-supervised rotation recognition task to a classifier $C : \tilde{\mathcal{X}} = \mathcal{T}(\mathcal{X}) \to \{1, 2, \cdots, K\}$ that shares parameters with the discriminator, and forced the generator to collaborate with the classifier on the rotation recognition task. Formally, the objective functions of SSGAN are given by:

$$\max_{D,C} V(G, D) + \lambda_d \cdot \mathbb{E}_{x \sim \mathcal{P}_d, T_k \sim \mathcal{T}}[\log C(k|T_k(x))], \tag{2}$$

$$\min_{G} V(G, D) - \lambda_g \cdot \mathbb{E}_{x \sim \mathcal{P}_g, T_k \sim \mathcal{T}}[\log C(k|T_k(x))], \tag{3}$$

where $\lambda_d$ and $\lambda_g$ are two hyper-parameters that balance the original GAN task and the additional self-supervised task, $\mathcal{T} = \{T_k\}_{k=1}^{K}$ is the set of deterministic data transformations such as rotations (i.e., $\mathcal{T} = \{0°, 90°, 180°, 270°\}$). Throughout this paper, data transformations are uniformly sampled by default (i.e., $p(T_k) = \frac{1}{K}, \forall T_k \in \mathcal{T}$), unless otherwise specified.

**Theorem 1** ([57]). *Given the optimal classifier* $C^*(k|\tilde{x}) = \frac{p_d^{T_k}(\tilde{x})}{\sum_{k=1}^{K} p_d^{T_k}(\tilde{x})}$ *of SSGAN, at the equilibrium point, maximizing the self-supervised task for the generator is equivalent to:*

$$\max_{G} \frac{1}{K} \sum_{k=1}^{K} \left[ \mathbb{E}_{\tilde{x} \sim \mathcal{P}_g^{T_k}} \log \left( \frac{p_d^{T_k}(\tilde{x})}{\sum_{k=1}^{K} p_d^{T_k}(\tilde{x})} \right) \right], \tag{4}$$

*where* $\mathcal{P}_g^{T_k}$, $\mathcal{P}_d^{T_k}$ *indicate the distribution of transformed generated or real data* $\tilde{x} \in \tilde{\mathcal{X}}$ *under the transformation* $T_k$ *with density of* $p_g^{T_k}(\tilde{x}) = \int \delta(\tilde{x} - T_k(x)) p_g(x) dx$ *or* $p_d^{T_k}(\tilde{x}) = \int \delta(\tilde{x} - T_k(x)) p_d(x) dx$.

All proofs of theorems/propositions including this one (for completeness) in this paper are referred to Appendix A. Theorem 1 reveals that the self-supervised task of SSGAN enforces the generator to produce only rotation-detectable images other than the whole real images [57], causing a goal inconsistent with generative modeling that faithfully learns the real data distribution. Consequently, SSGAN still requires the original GAN task, which forms a multi-task learning framework that needs hyper-parameters to trade off the two separate tasks.

## 2.3 SSGAN with Multi-Class Minimax Game

To improve the self-supervised task in SSGAN, SSGAN-MS [57] proposed a multi-class minimax self-supervised task by adding the fake class 0 to a label-extend classifier $C_+ : \tilde{\mathcal{X}} \to \{0, 1, 2, \cdots, K\}$, where the generator and the classifier compete on the self-supervised task. Formally, the objective functions of SSGAN-MS are defined as follows:

$$\max_{D,C_+} V(G, D) + \lambda_d \cdot (\mathbb{E}_{x \sim \mathcal{P}_d, T_k \sim \mathcal{T}}[\log C_+(k|T_k(x))] + \mathbb{E}_{x \sim \mathcal{P}_g, T_k \sim \mathcal{T}}[\log C_+(0|T_k(x))]), \tag{5}$$

$$\min_{G} V(G, D) - \lambda_g \cdot (\mathbb{E}_{x \sim \mathcal{P}_g, T_k \sim \mathcal{T}}[\log C_+(k|T_k(x))] - \mathbb{E}_{x \sim \mathcal{P}_g, T_k \sim \mathcal{T}}[\log C_+(0|T_k(x))]). \tag{6}$$

**Theorem 2.** *Given the optimal classifier* $C_+^*(k|\tilde{x}) = \frac{p_d^T(\tilde{x})}{p_g^T(\tilde{x})} \frac{p_d^{T_k}(\tilde{x})}{\sum_{k=1}^{K} p_d^{T_k}(\tilde{x})} C_+^*(0|\tilde{x})$ *of SSGAN-MS, at the equilibrium point, maximizing the self-supervised task for the generator is equivalent to*[1]:

$$\min_{G} \mathbb{D}_{\mathrm{KL}}(\mathcal{P}_g^T \| \mathcal{P}_d^T) - \frac{1}{K} \sum_{k=1}^{K} \left[ \mathbb{E}_{\tilde{x} \sim \mathcal{P}_g^{T_k}} \log \left( \frac{p_d^{T_k}(\tilde{x})}{\sum_{k=1}^{K} p_d^{T_k}(\tilde{x})} \right) \right], \tag{7}$$

*where* $\mathcal{P}_g^T$, $\mathcal{P}_d^T$ *represent the mixture distribution of transformed generated or real data* $\tilde{x} \in \tilde{\mathcal{X}}$ *with density of* $p_g^T(\tilde{x}) = \sum_{k=1}^{K} p(T_k) p_g^{T_k}(\tilde{x})$ *or* $p_d^T(\tilde{x}) = \sum_{k=1}^{K} p(T_k) p_d^{T_k}(\tilde{x})$.

---

[1]Note that our Theorem 2 corrects the wrong version in the SSGAN-MS paper [57], where the authors mistakenly regard $\frac{p_d^T(\tilde{x})}{p_g^T(\tilde{x})} = \frac{\sum_{k=1}^{K} p(T_k) p_d^{T_k}(\tilde{x})}{\sum_{k=1}^{K} p(T_k) p_g^{T_k}(\tilde{x})}$ as $\frac{p_d^{T_k}(\tilde{x})}{p_g^{T_k}(\tilde{x})}$ in their proof. Please see Appendix A.2 for details.

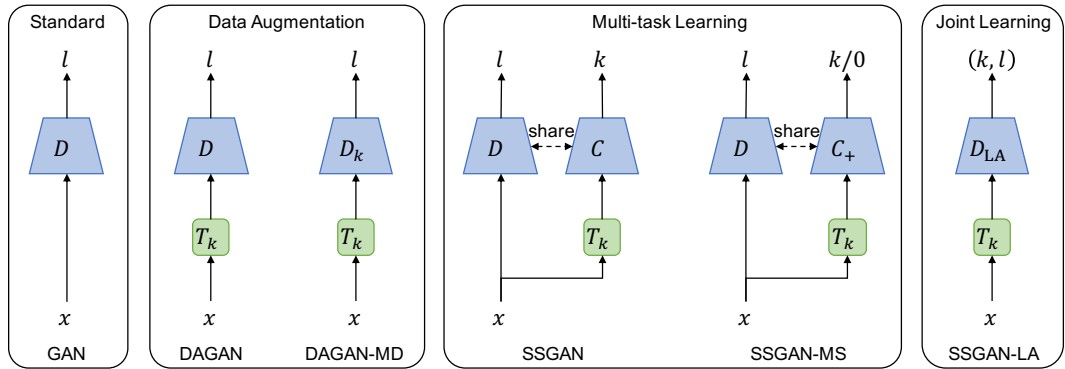

Figure 1: Schematics of the discriminators and classifiers of the competitive methods. We systematically divide these methods into four categories: Standard (GAN), Data Augmentation (DAGAN and DAGAN-MD), Multi-task Learning (SSGAN and SSGAN-MS), and Joint Learning (SSGAN-LA).

Our Theorem 2 states that the improved self-supervised task for the generator of SSGAN-MS, under the optimal classifier, retains the inconsistent goal in SSGAN that enforces the generator to produce typical rotation-detectable images rather than the entire real images. In addition, minimizing $\mathbb{D}_{\mathrm{KL}}(\mathcal{P}_g^T \| \mathcal{P}_d^T)$ cannot guarantee the generator to recover the real data distribution under the used data transformation setting (see Theorem 4). Due to the existence of these issues, SSGAN-MS also requires the original GAN task to approximate the real data distribution. We regard the learning paradigm of SSGAN-MS and SSGAN as the multi-task learning framework (see Figure 1).

## 3 Method

We note that the self-supervised classifiers of existing self-supervised GANs are agnostic to the density of the generator distribution $p_g(x)$ (or $p_g^{T_1}(\tilde{x})$) (see $C^*(k|\tilde{x})$ in Theorem 1 and $C_+^*(k|\tilde{x})$ in Theorem 2). In other words, their classifiers do not know how far away the current generator distribution $\mathcal{P}_g$ is from the data distribution $\mathcal{P}_d$, making them unable to provide any informative guidance to match the generator distribution with the data distribution. Consequently, existing self-supervised GANs suffer from the problem of involving an undesired goal for the generator when learning from the generator distribution-agnostic self-supervised classifiers. One naive solution for this problem is simply discarding the self-supervised task for the generator. However, the generator cannot take full advantage of the direct benefits of self-supervised learning in this case (see Appendix D).

In this work, our goal is to make full use of self-supervision to overcome catastrophic forgetting of the discriminator and enhance generation performance of the generator without introducing any undesired goal. We argue that the generator-agnostic issue of previous self-supervised classifiers originates from that those classifiers are trained to recognize the corresponding angles for only the rotated real images but not the rotated fake images. Motivated by the this understanding, we propose to allow the self-supervised classifier to discriminatively recognize the corresponding angles of the rotated real and fake images such that it can be aware of the generator distribution and the data distribution like the discriminator. In particular, the new classifier includes the role of the original discriminator, therefore the discriminator could be omitted though the convergence speed might be slowed down (see Appendix C for performance with the original discriminator). Specifically, we augment the original GAN labels (real or fake) via the self-supervision of data transformation to construct augmented labels and develop a novel label-augmented discriminator $D_{\mathrm{LA}} : \tilde{\mathcal{X}} \to \{1, 2, \cdots, K\} \times \{0, 1\}$ ($2K$ classes) to predict the augmented labels. By construction, this formulation forms a single multi-class classification task for the label-augmented discriminator, which is different from a single binary classification in standard GANs and two separate classification tasks in existing self-supervised GANs (see Figure 1). Formally, the objective function for the label-augmented discriminator of the proposed self-supervised GANs with label augmentation, dubbed SSGAN-LA, is defined as follows:

$$\max_{D_{\mathrm{LA}}} \mathbb{E}_{x \sim \mathcal{P}_d, T_k \sim \mathcal{T}}[\log D_{\mathrm{LA}}(k, 1|T_k(x))] + \mathbb{E}_{x \sim \mathcal{P}_g, T_k \sim \mathcal{T}}[\log D_{\mathrm{LA}}(k, 0|T_k(x))], \tag{8}$$

where $D_{\mathrm{LA}}(k, 1|T_k(x))$ (resp. $D_{\mathrm{LA}}(k, 0|T_k(x))$) outputs the probability that a transformed data $T_k(x)$ is jointly classified as the augmented label of real (resp. fake) and the $k$-th transformation.

**Proposition 1.** *For any fixed generator, given a data $\tilde{x} \in \tilde{\mathcal{X}}$ that drawn from mixture distribution of transformed data, the optimal label-augmented discriminator of SSGAN-LA has the form of:*

$$D_{\text{LA}}^*(k, 1|\tilde{x}) = \frac{p_d^{T_k}(\tilde{x})}{\sum_{k=1}^K (p_d^{T_k}(\tilde{x}) + p_g^{T_k}(\tilde{x}))}, D_{\text{LA}}^*(k, 0|\tilde{x}) = \frac{p_g^{T_k}(\tilde{x})}{\sum_{k=1}^K (p_d^{T_k}(\tilde{x}) + p_g^{T_k}(\tilde{x}))}. \quad (9)$$

Proposition 1 shows that the optimal label-augmented discriminator of SSGAN-LA is able to aware the densities of both real and generated data distributions under every transformation (i.e., $p_d^{T_k}(\tilde{x})$ and $p_g^{T_k}(\tilde{x})$), and thus can provide the discrepancy (i.e., $p_d^{T_k}(\tilde{x})/p_g^{T_k}(\tilde{x}) = D^*(k, 1|\tilde{x})/D^*(k, 0|\tilde{x})$) between the data distribution and the generator distribution under every transformation to optimize the generator. Consequently, we now propose to optimize the generator of SSGAN-LA by optimizing such discrepancy using the objective function formulated as follows:

$$\max_G \mathbb{E}_{x \sim \mathcal{P}_g, T_k \sim \mathcal{T}}[\log D_{\text{LA}}(k, 1|T_k(x))] - \mathbb{E}_{x \sim \mathcal{P}_g, T_k \sim \mathcal{T}}[\log D_{\text{LA}}(k, 0|T_k(x))]. \quad (10)$$

**Theorem 3.** *The objective function for the generator of SSGAN-LA, given the optimal label-augmented discriminator, boils down to:*

$$\min_G \frac{1}{K} \sum_{k=1}^K \mathbb{D}_{\text{KL}}(\mathcal{P}_g^{T_k} \| \mathcal{P}_d^{T_k}). \quad (11)$$

*The global minimum is achieved if and only if $\mathcal{P}_g = \mathcal{P}_d$ when $\exists T_k \in \mathcal{T}$ is an invertible transformation.*

Theorem 3 shows that the generator of SSGAN-LA is encouraged to approximate the real data distribution under different transformations measured by the reverse KL divergence without any goal inconsistent with generative modeling, unlike previous self-supervised GANs. In particular, the generator is accordingly guaranteed to replicate the real data distribution when an existing transformation is invertible such as the identity and 90-degree rotation transformations. In summary, SSGAN-LA seamlessly incorporates self-supervised learning into the original GAN objective to improve the training stability on the premise of faithfully learning of the real data distribution.

Compared with standard GANs, the transformation prediction task, especially on the real samples, establishes stationary self-supervised learning environments in SSGAN-LA to prevent the discriminator from catastrophic forgetting. In addition, the multiple discriminative signals under different transformations are capable of providing diverse feedback to obtain a well-performed generator. Compared with previous self-supervised GANs, in addition to the unbiased learning objective for the generator, the discriminator of SSGAN-LA has the ability to learn better representations. In particular, the discriminator utilizes not only the real data but also the generated data as augmented data to obtain robust transformation-recognition ability, and distinguishes between real data and generated data from multiple perspectives via different transformations to obtain enhanced discrimination ability. These two abilities enable the discriminator to learn more meaningful representations than existing self-supervised GANs, which could in turn promote the generation performance of the generator.

## 4 The Importance of Self-Supervision

The proposed SSGAN-LA might benefit from two potential aspects, i.e., augmented data and the corresponding self-supervision. To verify the importance of the latter, we introduce two ablation models that do not explicitly receive feedback from the self-supervision as competitors.

### 4.1 Data Augmentation GAN

We first consider a naive data augmentation GAN (DAGAN) that completely ignores self-supervision. Formally, the objective function of DAGAN is formulated as follows:

$$\min_G \max_D \mathbb{E}_{x \sim \mathcal{P}_d, T_k \sim \mathcal{T}}[\log D(1|T_k(x))] + \mathbb{E}_{x \sim \mathcal{P}_g, T_k \sim \mathcal{T}}[\log D(0|T_k(x))]. \quad (12)$$

This formulation is mathematically equivalent to DiffAugment [66] and IDA [59]. We regard this approach as an ablation model of SSGAN-LA as it trains with only the GAN labels but ignores the self-supervised labels. Next, we show the indispensability of self-supervision.

**Theorem 4.** *At the equilibrium point of DAGAN, the optimal generator implies $\mathcal{P}_g^T = \mathcal{P}_d^T$. However, if $(\mathcal{T}, \circ)$ forms a group and $T_k \in \mathcal{T}$ is uniformly sampled, then the probability that the optimal generator replicates the real data distribution is $\mathbb{P}(\mathcal{P}_g = \mathcal{P}_d | \mathcal{P}_g^T = \mathcal{P}_d^T) = 0$.*

Theorem 4 suggests that the formulation of DAGAN is insufficient to recover the real data distribution for the generator as $(\{0°, 90°, 180°, 270°\}, \circ)$ forms a cyclic group and the transformation is uniformly sampled by default. Intuitively, if transformations are not identified by their corresponding self-supervision and can be represented by others, then the transformed data will be indistinguishable from the original data so that they may be leaked to the generator. In more detail, the generator would converge to an arbitrary mixture distribution of transformed real data (see Appendix A.5). Note that all theoretical results in this paper are not limited to rotation but many other types of data transformation, e.g., RBG permutation and patch shuffling. Nonetheless, we follow the practice of SSGAN to use rotation as the data transformation for fairness. In summary, data augmentation GANs may even suffer from the convergence problem without utilizing self-supervision.

### 4.2 DAGAN with Multiple Discriminators

To verify the importance of self-supervision while eliminating the convergence problem of DAGAN, we introduce another data augmentation GAN that has the same convergence point as SSGAN-LA. DAG [59] proposed a data augmentation GAN with multiple discriminators $\{D_k\}_{k=1}^K$ (we refer it as DAGAN-MD throughout this paper), which are indexed by the self-supervision and share all layers but the head with each other, to solve the convergence issue of DAGAN. Formally, the objective function of DAGAN-MD is formulated as:

$$\min_G \max_{\{D_k\}_{k=1}^K} \mathbb{E}_{x \sim \mathcal{P}_d, T_k \sim \mathcal{T}}[\log D_k(1|T_k(x))] + \mathbb{E}_{x \sim \mathcal{P}_g, T_k \sim \mathcal{T}}[\log D_k(0|T_k(x))]. \tag{13}$$

We regard DAGAN-MD as an ablation model of SSGAN-LA since DAGAN-MD takes the self-supervised signal as input while SSGAN-LA views it as target (see Figure 1). It is proved that the generator of DAGAN-MD could approximate the real data distribution at the optimum [59]. However, the lack of leveraging self-supervised signals as supervision will disadvantage the performance of DAGAN-MD in two ways. On one hand, the discriminators cannot learn stable representations to resist catastrophic forgetting without learning from self-supervised tasks. On the other hand, the discriminators cannot learn high-level representations contained in the specific semantics of different transformations. Specifically, these discriminators presumably enforce invariance to the shared layers [28], which could hurt the performance of representation learning as transformations modify the semantics of data. As we emphasized above, the more useful information learned by the discriminator, the better guidance provided to the generator, and the better optimization result reached by the generator. Accordingly, the performance of generative modeling would be affected.

## 5 Experiments

Our code is available at https://github.com/houliangict/ssgan-la.

### 5.1 SSGAN-LA Faithfully Learns the Real Data Distribution

We experiment on a synthetic dataset to intuitively verify whether SSGAN, SSGAM-MS, and SSGAN-LA can accurately match the real data distribution. The real data distribution is a one-dimensional normal distribution $\mathcal{P}_d = \mathcal{N}(0, 1)$. The set of transformations is $\mathcal{T} = \{T_k\}_{k=1}^{K=4}$ with the $k$-th transformation of $T_k(x) := x + 2(k-1)$ ($T_1$ is the identity transformation). We assume that the distributions of real data through different transformations have non-negligible overlaps. The generator and discriminator networks are implemented by multi-layer perceptrons with hidden size of 10 and non-linearity of Tanh. Figure 2 shows the density estimation of the target and generated data estimated by kernel density estimation [44] through sampling 10k data. SSGAN learns a biased distribution as its self-supervised task forces the generator to produce transformation-classifiable data. SSGAN-MS slightly mitigates this issue, but still cannot accurately approximate the real data distribution. The proposed SSGAN-LA successfully replicates the data distribution and achieves the best maximum mean discrepancy (MMD) [15] results as reported in brackets. In general, experimental results on synthetic data accord well with theoretical results of self-supervised methods.

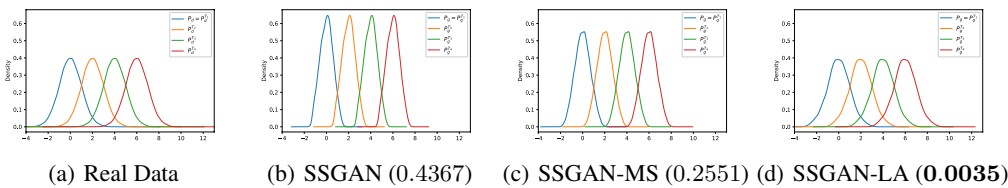

| (a) Real Data | (b) SSGAN (0.4367) | (c) SSGAN-MS (0.2551) | (d) SSGAN-LA (**0.0035**) |

Figure 2: Learned distribution on one-dimensional synthetic data. The blue line at left-most is the untransformed real or generated data distribution. The numbers in brackets are the MMD [15] scores.

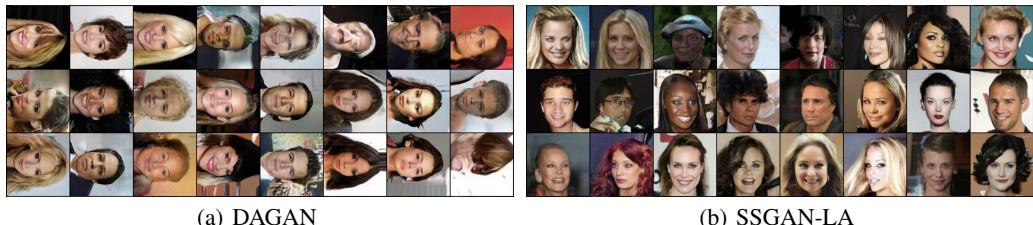

| (a) DAGAN | (b) SSGAN-LA |

Figure 3: Generated samples on CelebA. DAGAN suffers from the augmentation-leaking issue.

## 5.2 SSGAN-LA Resolves the Augmentation-Leaking Issue

Figure 3 shows images randomly generated by DAGAN and SSGAN-LA on the CelebA dataset [34]. Apparently, the generator of DAGAN converges to generate the rotated images, validating Theorem 4 that DAGAN cannot guarantee to recover the original data distribution under the used rotation-based data augmentation setting. The proposed SSGAN-LA resolves this augmentation-leaking problem attributed to explicitly identifying different rotated images by predicting the self-supervised pseudo signals (i.e., rotation angles).

## 5.3 Comparison of Sample Quality

We conduct experiments on three real-world datasets: CIFAR-10 [26], STL-10 [8], and Tiny-ImageNet [27]. We implement all methods based on unconditional BigGAN [4] without accessing human-annotated labels. We defer the detailed experimental settings into Appendix B due to the limited space. To evaluate the performance of methods on sample quality, we adopt two widely-used evaluation metrics: Fréchet Inception Distance (FID) [17] and Inception Score (IS) [49]. We follow the practice of BigGAN to randomly sample 50k images to calculate the IS and FID scores.

As shown in Table 1, SSGAN and SSGAN-MS surpass GAN due to the auxiliary self-supervised tasks that encourage discriminators to learn stable representations to resist the forgetting issue. As DAGAN suffers from the augmentation-leaking problem [22] in the default setting, we up-weight the probability of the identity transformation for DAGAN (named DAGAN$^+$) to avoid the undesirable augmentation-leaking problem. DAGAN$^+$ and DAGAN-MD have made considerable improvements over GAN on CIFAR-10 and STL-10 but a substantial degradation on Tiny-ImageNet, indicating

Table 1: FID (↓) and IS (↑) comparison of methods on CIFAR-10, STL-10, and Tiny-ImageNet.

| Dataset | Metric | GAN | SSGAN | SSGAN-MS | DAGAN$^+$ | DAGAN-MD | SSGAN-LA |
|---------|--------|-----|-------|----------|-----------|----------|----------|
| CIFAR-10 | FID (↓) | 10.83 | 7.52 | 7.08 | 10.01 | 7.80 | **5.87** |
|  | IS (↑) | 8.42 | 8.29 | 8.45 | 8.10 | 8.31 | **8.57** |
| STL-10 | FID (↓) | 20.15 | 16.84 | 16.46 | 16.42 | 16.16 | **14.58** |
|  | IS (↑) | 10.25 | 10.36 | 10.40 | 10.27 | 10.31 | **10.61** |
| Tiny-ImageNet | FID (↓) | 31.01 | 30.09 | 25.76 | 52.54 | 39.57 | **23.69** |
|  | IS (↑) | 9.80 | 10.21 | 10.57 | 7.55 | 8.44 | **11.14** |

Table 2: Accuracy (↑) comparison of methods on CIFAR-10, CIFAR-100, and Tiny-ImageNet based on learned representations extracted from each residual block of the discriminator.

| Dataset | Block | GAN | SSGAN | SSGAN-MS | DAGAN$^+$ | DAGAN-MD | SSGAN-LA |
|---|---|---|---|---|---|---|---|
| CIFAR-10 | Block1 | **0.641** | 0.636 | 0.640 | 0.639 | 0.633 | 0.637 |
| | Block2 | 0.729 | 0.722 | 0.728 | 0.726 | 0.720 | **0.741** |
| | Block3 | 0.765 | 0.769 | 0.770 | 0.764 | 0.746 | **0.782** |
| | Block4 | 0.769 | 0.784 | 0.787 | 0.772 | 0.768 | **0.803** |
| CIFAR-100 | Block1 | 0.323 | 0.320 | 0.322 | 0.323 | **0.324** | 0.318 |
| | Block2 | 0.456 | 0.452 | 0.461 | 0.460 | 0.443 | **0.468** |
| | Block3 | 0.494 | 0.493 | 0.502 | 0.493 | 0.478 | **0.511** |
| | Block4 | 0.504 | 0.513 | 0.521 | 0.519 | 0.492 | **0.543** |
| Tiny-ImageNet | Block1 | 0.142 | 0.138 | 0.132 | **0.155** | 0.135 | 0.129 |
| | Block2 | 0.221 | **0.224** | 0.214 | 0.184 | 0.184 | 0.209 |
| | Block3 | 0.251 | 0.277 | 0.280 | 0.211 | 0.231 | **0.283** |
| | Block4 | 0.325 | 0.346 | 0.344 | 0.263 | 0.279 | **0.349** |
| | Block5 | 0.141 | 0.313 | 0.341 | 0.110 | 0.224 | **0.351** |

that only data augmentation without self-supervised signals cannot bring consistent improvements. The proposed SSGAN-LA significantly outperforms existing SSGANs and DAGANs by inheriting their advantages while overcoming their shortcomings. Compared with existing SSGANs, SSGAN-LA inherits the benefits of self-supervised tasks while avoiding the undesired learning objective. Compared with DAGANs, SSGAN-LA inherits the benefits of augmented data and utilizes self-supervised signals to enhance the representation learning ability of the discriminator that could provide more valuable guidance to the generator.

## 5.4 Comparison of Representation Quality

In order to check whether the discriminator learns meaningful representations, we train a 10-way logistic regression classifier on CIFAR-10, 100-way on CIFAR-100, and 200-way on Tiny-ImageNet, respectively, using the learned representation of real data extracted from residual blocks in the discriminator. Specifically, we train the classifier with a batch size of 128 for 50 epochs. The optimizer is Adam with a learning rate of 0.05 and decayed by 10 at both epoch 30 and epoch 40, following the practice of [6]. The linear model is trained on the training set and tested on the validation set of the corresponding datasets. Classification accuracy is the evaluation metric.

As shown in Table 2, SSGAN and SSGAN-MS obtain improvements on GAN, confirming that the self-supervised tasks facilitate learning more meaningful representations. DAGAN-MD performs substantially worse than the self-supervised counterparts and even worse than GAN and DAGAN$^+$ in some cases, validating that the multiple discriminators of DAGAN-MD tend to weaken the representation learning ability in the shared blocks. In other words, the shared blocks are limited to invariant features, which may lose useful information about data. Compared with all competitive baselines, the proposed SSGAN-LA achieves the best accuracy results on most blocks, especially the deep blocks. These results verify that the discriminator of SSGAN-LA learns meaningful high-level representations. We argue that the powerful representation ability of the discriminator could in turn promote the performance of generative modeling of the generator.

## 5.5 SSGAN-LA Overcomes Catastrophic Forgetting

Figure 4 plots the accuracy (calculated following Section 5.4) curves of different methods during GAN training on CIFAR-10. With the increase of GAN training iterations, the accuracy results of GAN and DAGAN-MD first increased and then decreased, indicating that their discriminators both suffer from catastrophic forgetting to some extent. The proposed SSGAN-LA achieves consecutive increases in terms of accuracy and consistently surpasses other competitive methods, verifying that utilizing self-supervision can effectively overcome the catastrophic forgetting issue.

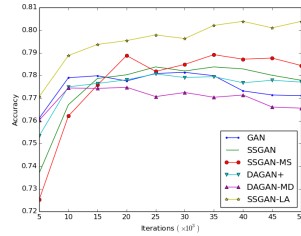

Figure 4: Accuracy curves during GAN training on CIFAR-10.

Table 3: FID and IS comparison under full and limited data regimes on CIFAR-10 and CIFAR-100.

| Metric | Method | CIFAR-10 | | | CIFAR-100 | | |
|---|---|---|---|---|---|---|---|
| | | 100% | 20% | 10% | 100% | 20% | 10% |
| FID | BigGAN [4] + DiffAugment [66] | 8.70 | 14.04 | 22.40 | 12.00 | 22.14 | 33.70 |
| | + SSGAN-LA | **8.05** | **11.16** | **15.08** | **9.77** | **15.91** | **23.17** |
| IS | BigGAN [4] + DiffAugment [66] | 9.16 | 8.65 | 8.09 | 10.66 | 9.47 | 8.38 |
| | + SSGAN-LA | **9.30** | **8.84** | **8.30** | **11.13** | **10.52** | **9.81** |

### 5.6 SSGAN-LA Improves the Data Efficiency of GANs

In this section, we experiment on the full and limited data regimes on CIFAR-10 and CIFAR-100 compared with the state-of-the-art data augmentation method (DiffAugment [66]) for training GANs. We adopt the codebase of DiffAugment and follow its practice to randomly sample 10k data for calculating the FID and IS scores for a fair comparison. As shown in Table 3, SSGAN-LA significantly improves the data efficiency of GANs and to our knowledge achieves the new state-of-the-art FID and IS results under the full and limited data regimes on CIFAR-10 and CIFAR-100 based on the BigGAN [4] backbone. We argue that the reason is that the proposed label-augmented discriminator is less easy to over-fitting than the normal discriminator by solving a more challenging task.

## 6 Related Work

Since generative adversarial networks (GANs) [14] appearance, numerous variants focus on improving the generation performance from perspectives of network architecture [48, 63, 5, 4, 23, 24], divergence minimization [43, 2, 16, 30, 53, 21], learning procedure [37, 51, 18], weight normalization [38, 50, 67], model regularization [46, 62, 54], and multiple discriminators [41, 12, 40, 10, 1]. Recently, self-supervision has been applied to GANs to improve the training stability. Self-supervised learning is a family of unsupervised representation learning methods that solve pretext tasks. Various self-supervised pretext tasks on image representation learning have been proposed, e.g., predicting context [11], solving jigsaw puzzle [42], and recognizing rotation [13]. [6] introduced an auxiliary rotation prediction task [13] onto the discriminator to alleviate catastrophic forgetting. [57, 58] analyzed the drawback of the self-supervised task in [6] and proposed an improved self-supervised task. [35] combined self- and semi-supervised learning to surpass fully supervised GANs. [3] introduced a deshuffling task to the discriminator that solves the jigsaw puzzle. [19] required the discriminator to examine the structural consistency of real images. [45] extended the self-supervision into the latent space by detecting the latent transformation. [29] explored contrastive learning and mutual information maximization to simultaneously tackle the catastrophic forgetting and mode collapse issues. [64] presented a transformation-based consistency regularization, which has improved in [68, 69]. The similar label augmentation technique [28] for discriminators is also proposed in DADA [65] and [60]. However, the former (DADA) aimed at data augmentation for few-shot classification, and the latter focused on self-labeling in semi-supervised learning. Most importantly, they are both different from ours with the loss function for the generator, which is the key for the Theorem 3 that guarantees the generator to faithfully learn the real data distribution.

## 7 Conclusion

In this paper, we present a novel formulation of transformation-based self-supervised GANs that is able to faithfully learn the real data distribution. Specifically, we unify the original GAN task with the self-supervised task into a single multi-class classification task for the discriminator (forming a label-augmented discriminator) by augmenting the original GAN labels (real or fake) via self-supervision of data transformation. Consequently, the generator is encouraged to minimize the reversed Kullback–Leibler divergence between the real data distribution and the generated data distribution under different transformations provided by the optimal label-augmented discriminator. We provide both theoretical analysis and empirical evidence to support the superiority of the proposed method compared with previous self-supervised GANs and data augmentation GANs. In the future, we will explore and exploit the proposed approach to semi- and fully supervised GANs.

## Acknowledgments and Disclosure of Funding

This work is funded by the National Natural Science Foundation of China under Grant Nos. 62102402, 91746301, and National Key R&D Program of China (2020AAA0105200). Huawei Shen is also supported by Beijing Academy of Artificial Intelligence (BAAI) under the grant number BAAI2019QN0304 and K.C. Wong Education Foundation.

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
