# A   Proofs

In order to mathematically analyze transformation-based self-supervised (or data augmentation) GANs, we need to rewrite the objective functions that are easy to calculate derivatives. Considering that a data $x \in \mathcal{X}$ and a transformation $T_k \in \mathcal{T}$ are independent from each other and the transformed data $\tilde{x} = T_k(x) \in \tilde{\mathcal{X}} = \mathcal{T}(\mathcal{X})$ is deterministic depended on both $x$ and $T_k$, the density of the joint distribution is $p(\tilde{x}, x, T_k) = p(x)p(T_k)p(\tilde{x}|x, T_k) = p(x)p(T_k)\delta(\tilde{x} - T_k(x))$ with the indicator function $\delta(0) = 1$ and $\delta(\tilde{x}) = 0, \forall \tilde{x} \neq 0$.

**Proposition 2.** *For any continuous and differentiable function $f$ whose domain is $\tilde{\mathcal{X}}$, we have:*

$$\mathbb{E}_{x \sim \mathcal{P}, T_k \sim \mathcal{T}}[\log f(T_k(x))] = \mathbb{E}_{\tilde{x} \sim \mathcal{P}^T, T_k \sim \mathcal{T}^{\tilde{x}}}[\log f(\tilde{x})] = \mathbb{E}_{T_k \sim \mathcal{T}, \tilde{x} \sim \mathcal{P}^{T_k}}[\log f(\tilde{x})], \quad (14)$$

*where $\mathcal{P}$ denotes the original data distribution, $\mathcal{P}^T$ indicates the mixture distribution of transformed data, $\mathcal{P}^{T_k}$ means the distribution of transformed data given the transformation $T_k$, and $\mathcal{T}^{\tilde{x}}$ represents the distribution of transformation given the transformed data $\tilde{x}$.*

*Proof.*

$$\mathbb{E}_{x \sim P, T_k \sim \mathcal{T}}[\log f(T_k(x))] \tag{15}$$

$$= \int p(x) \sum_{k=1}^{K} p(T_k) \log f(T_k(x)) dx \tag{16}$$

$$= \int p(x) \sum_{k=1}^{K} p(T_k)\delta(\tilde{x} - T_k(x)) \log f(\tilde{x}) dx d\tilde{x} \tag{17}$$

$$= \int p(x) \sum_{k=1}^{K} p(T_k)p(\tilde{x}|x, T_k) \log f(\tilde{x}) dx d\tilde{x} \tag{18}$$

$$= \int \sum_{k=1}^{K} p(\tilde{x}, x, T_k) \log f(\tilde{x}) dx d\tilde{x} \tag{19}$$

$$= \int \sum_{k=1}^{K} p(\tilde{x}, T_k) \log f(\tilde{x}) d\tilde{x} \tag{20}$$

$$= \int p^T(\tilde{x}) \sum_{k=1}^{K} p(T_k|\tilde{x}) \log f(\tilde{x}) d\tilde{x} = \mathbb{E}_{\tilde{x} \sim P^T, T_k \sim \mathcal{T}^{\tilde{x}}}[\log f(\tilde{x})] \tag{21}$$

$$= \sum_{k=1}^{K} \int p(T_k)p(\tilde{x}|T_k) \log f(\tilde{x}) d\tilde{x} = \mathbb{E}_{T_k \sim \mathcal{T}, \tilde{x} \sim \mathcal{P}^{T_k}}[\log f(\tilde{x})]. \tag{22}$$

□

## A.1   Proof of Theorem 1

Theorem 1 was proved in the SSGAN-MS paper [57]. We here give a brief proof for completeness of this paper. Readers are encouraged to refer to the original proof in [57] for more details.

**Theorem 1** ([57]). *Given the optimal classifier $C^*(k|\tilde{x}) = \frac{p_d^{T_k}(\tilde{x})}{\sum_{k=1}^{K} p_d^{T_k}(\tilde{x})}$ of SSGAN, at the equilibrium point, maximizing the self-supervised task for the generator is equivalent to:*

$$\max_{G} \frac{1}{K} \sum_{k=1}^{K} \left[ \mathbb{E}_{\tilde{x} \sim \mathcal{P}_g^{T_k}} \log \left( \frac{p_d^{T_k}(\tilde{x})}{\sum_{k=1}^{K} p_d^{T_k}(\tilde{x})} \right) \right], \tag{4}$$

*where $\mathcal{P}_g^{T_k}$, $\mathcal{P}_d^{T_k}$ indicate the distribution of transformed generated or real data $\tilde{x} \in \tilde{\mathcal{X}}$ under the transformation $T_k$ with density of $p_g^{T_k}(\tilde{x}) = \int \delta(\tilde{x} - T_k(x))p_g(x)dx$ or $p_d^{T_k}(\tilde{x}) = \int \delta(\tilde{x} - T_k(x))p_d(x)dx$.*

*Proof.* According to Proposition 2, the objective function of the self-supervised task for the classifier of SSGAN can be rewritten as follows:

$$\max_C \mathbb{E}_{x \sim \mathcal{P}_d, T_k \sim \mathcal{T}}[\log C(k|T_k(x))] \Rightarrow \max_C \mathbb{E}_{\tilde{x} \sim \mathcal{P}_d^T, T_k \sim \mathcal{T}_d^{\tilde{x}}}[\log C(k|\tilde{x})]. \tag{23}$$

According to the Proposition 1 in [57], the optimal classifier $C^*$ has the form of:

$$C^*(k|\tilde{x}) = \frac{p_d^{T_k}(\tilde{x})}{\sum_{k=1}^K p_d^{T_k}(\tilde{x})}. \tag{24}$$

Therefore, the objective function of the self-supervised task for the generator of SSGAN, under the optimal classifier, can be considered as the following objective:

$$\max_G \mathbb{E}_{x \sim \mathcal{P}_g, T_k \sim \mathcal{T}}[\log C^*(k|T_k(x))] \tag{25}$$

$$\Rightarrow \max_G \mathbb{E}_{T_k \sim \mathcal{T}, \tilde{x} \sim \mathcal{P}_g^{T_k}}[\log C^*(k|\tilde{x})] \tag{26}$$

$$\Rightarrow \max_G \frac{1}{K} \sum_{k=1}^K \left[ \mathbb{E}_{\tilde{x} \sim \mathcal{P}_g^{T_k}} \log \left( \frac{p_d^{T_k}(\tilde{x})}{\sum_{k=1}^K p_d^{T_k}(\tilde{x})} \right) \right]. \tag{27}$$

$\square$

## A.2  Proof of Theorem 2

**Theorem 2.** *Given the optimal classifier $C_+^*(k|\tilde{x}) = \frac{p_d^T(\tilde{x})}{p_g^T(\tilde{x})} \frac{p_d^{T_k}(\tilde{x})}{\sum_{k=1}^K p_d^{T_k}(\tilde{x})} C_+^*(0|\tilde{x})$ of SSGAN-MS, at the equilibrium point, maximizing the self-supervised task for the generator is equivalent to[2]:*

$$\min_G \mathbb{D}_{\mathrm{KL}}(\mathcal{P}_g^T \| \mathcal{P}_d^T) - \frac{1}{K} \sum_{k=1}^K \left[ \mathbb{E}_{\tilde{x} \sim \mathcal{P}_g^{T_k}} \log \left( \frac{p_d^{T_k}(\tilde{x})}{\sum_{k=1}^K p_d^{T_k}(\tilde{x})} \right) \right], \tag{7}$$

*where $\mathcal{P}_g^T$, $\mathcal{P}_d^T$ represent the mixture distribution of transformed generated or real data $\tilde{x} \in \tilde{\mathcal{X}}$ with density of $p_g^T(\tilde{x}) = \sum_{k=1}^K p(T_k) p_g^{T_k}(\tilde{x})$ or $p_d^T(\tilde{x}) = \sum_{k=1}^K p(T_k) p_d^{T_k}(\tilde{x})$.*

*Proof.* According to Proposition 2, we first rewrite the objective function of the self-supervised task for the classifier of SSGAN-MS as follows:

$$\max_{C_+} \mathbb{E}_{x \sim \mathcal{P}_d, T_k \sim \mathcal{T}}[\log C_+(k|T_k(x))] + \mathbb{E}_{x \sim \mathcal{P}_g, T_k \sim \mathcal{T}}[\log C_+(0|T_k(x))] \tag{28}$$

$$\Rightarrow \max_{C_+} \mathbb{E}_{\tilde{x} \sim \mathcal{P}_d^T, T_k \sim \mathcal{T}_d^{\tilde{x}}}[\log C_+(k|\tilde{x})] + \mathbb{E}_{\tilde{x} \sim \mathcal{P}_g^T, T_k \sim \mathcal{T}_g^{\tilde{x}}}[\log C_+(0|\tilde{x})]. \tag{29}$$

According to the Proposition 2 of [57], for any fixed generator, the optimal classifier $C_+^*$ is:

$$C_+^*(k|\tilde{x}) = \frac{p_d^T(\tilde{x})}{p_g^T(\tilde{x})} \frac{p_d^{T_k}(\tilde{x})}{\sum_{k=1}^K p_d^{T_k}(\tilde{x})} C_+^*(0|\tilde{x}), \forall k \in \{1, 2, \cdots, K\}. \tag{30}$$

Since $\sum_{k=0}^K C_+^*(k|\tilde{x}) = 1$ for each transformed data $\tilde{x} \in \tilde{\mathcal{X}}$, we have:

$$C_+^*(0|\tilde{x}) = \frac{p_g^T(\tilde{x})}{p_d^T(\tilde{x}) + p_g^T(\tilde{x})}, \tag{31}$$

$$C_+^*(k|\tilde{x}) = \frac{p_d^T(\tilde{x})}{p_d^T(\tilde{x}) + p_g^T(\tilde{x})} \frac{p_d^{T_k}(\tilde{x})}{\sum_{k=1}^K p_d^{T_k}(\tilde{x})}, \forall k \in \{1, 2, \cdots, K\}. \tag{32}$$

---

[2]Note that our Theorem 2 corrects the wrong version in the SSGAN-MS paper [57], where the authors mistakenly regard $\frac{p_d^T(\tilde{x})}{p_g^T(\tilde{x})} = \frac{\sum_{k=1}^K p(T_k) p_d^{T_k}(\tilde{x})}{\sum_{k=1}^K p(T_k) p_g^{T_k}(\tilde{x})}$ as $\frac{p_d^{T_k}(\tilde{x})}{p_g^{T_k}(\tilde{x})}$ in their proof. Please see Appendix A.2 for details.

The self-supervised task for the generator of SSGAN-MS, under the optimal classifier, is equal to:

$$\max_G \mathbb{E}_{x \sim \mathcal{P}_g, T_k \sim \mathcal{T}}[\log C_+^*(k|T_k(x))] - \mathbb{E}_{x \sim \mathcal{P}_g, T_k \sim \mathcal{T}}[\log C_+^*(0|T_k(x))] \tag{33}$$

$$\Rightarrow \max_G \mathbb{E}_{\tilde{x} \sim \mathcal{P}_g^T, T_k \sim \mathcal{T}_g^{\tilde{x}}}[\log C_+^*(k|\tilde{x})] - \mathbb{E}_{\tilde{x} \sim \mathcal{P}_g^T, T_k \sim \mathcal{T}_g^{\tilde{x}}}[\log C_+^*(0|\tilde{x})] \tag{34}$$

$$\Rightarrow \max_G \mathbb{E}_{\tilde{x} \sim \mathcal{P}_g^T, T_k \sim \mathcal{T}_g^{\tilde{x}}}\left[\log \frac{C_+^*(k|\tilde{x})}{C_+^*(0|\tilde{x})}\right] \tag{35}$$

$$\Rightarrow \max_G \int p_g^T(\tilde{x}) \sum_{k=1}^{K} p_g(T_k|\tilde{x}) \log \left(\frac{p_d^T(\tilde{x})}{p_g^T(\tilde{x})} \frac{p_d^{T_k}(\tilde{x})}{\sum_{k=1}^{K} p_d^{T_k}(\tilde{x})}\right) d\tilde{x} \tag{36}$$

$$\Rightarrow \max_G \int p_g^T(\tilde{x}) \log \left(\frac{p_d^T(\tilde{x})}{p_g^T(\tilde{x})}\right) d\tilde{x} + \int p_g^T(\tilde{x}) \sum_{k=1}^{K} p_g(T_K|\tilde{x}) \log \left(\frac{p_d^{T_k}(\tilde{x})}{\sum_{k=1}^{K} p_d^{T_k}(\tilde{x})}\right) d\tilde{x} \tag{37}$$

$$\Rightarrow \max_G \int p_g^T(\tilde{x}) \log \left(\frac{p_d^T(\tilde{x})}{p_g^T(\tilde{x})}\right) d\tilde{x} + \sum_{k=1}^{K} p(T_K) \int p_g^{T_k}(\tilde{x}) \log \left(\frac{p_d^{T_k}(\tilde{x})}{\sum_{k=1}^{K} p_d^{T_k}(\tilde{x})}\right) d\tilde{x} \tag{38}$$

$$\Rightarrow \min_G \mathbb{D}_{\mathrm{KL}}(\mathcal{P}_g^T \| \mathcal{P}_d^T) - \frac{1}{K} \sum_{k=1}^{K} \left[\mathbb{E}_{\tilde{x} \sim P_g^{T_k}} \log \left(\frac{p_d^{T_k}(\tilde{x})}{\sum_{k=1}^{K} p_d^{T_k}(\tilde{x})}\right)\right]. \tag{39}$$

$$\square$$

## A.3 Proof of Proposition 1

**Proposition 1.** *For any fixed generator, given a data $\tilde{x} \in \tilde{\mathcal{X}}$ that drawn from mixture distribution of transformed data, the optimal label-augmented discriminator of SSGAN-LA has the form of:*

$$D_{\mathrm{LA}}^*(k, 1|\tilde{x}) = \frac{p_d^{T_k}(\tilde{x})}{\sum_{k=1}^{K}(p_d^{T_k}(\tilde{x}) + p_g^{T_k}(\tilde{x}))}, D_{\mathrm{LA}}^*(k, 0|\tilde{x}) = \frac{p_g^{T_k}(\tilde{x})}{\sum_{k=1}^{K}(p_d^{T_k}(\tilde{x}) + p_g^{T_k}(\tilde{x}))}. \tag{9}$$

*Proof.* We can first rewrite the objective function for the label-augmented discriminator as follows:

$$\max_{D_{\mathrm{LA}}} \mathbb{E}_{x \sim \mathcal{P}_d, T_k \sim \mathcal{T}}[\log D_{\mathrm{LA}}(k, 1|T_k(x))] + \mathbb{E}_{x \sim \mathcal{P}_g, T_k \sim \mathcal{T}}[\log D_{\mathrm{LA}}(k, 0|T_k(x))] \tag{40}$$

$$\Rightarrow \max_{D_{\mathrm{LA}}} \mathbb{E}_{\tilde{x} \sim \mathcal{P}_d^T, T_k \sim \mathcal{T}_d^{\tilde{x}}}[\log D_{\mathrm{LA}}(k, 1|\tilde{x})] + \mathbb{E}_{\tilde{x} \sim \mathcal{P}_g^T, T_k \sim \mathcal{T}_g^{\tilde{x}}}[\log D_{\mathrm{LA}}(k, 0|\tilde{x})] \tag{41}$$

$$\Rightarrow \max_{D_{\mathrm{LA}}} \int p_d^T(\tilde{x}) \sum_{k=1}^{K} p_d(T_k|\tilde{x}) \log D_{\mathrm{LA}}(k, 1|\tilde{x}) + p_g^T(\tilde{x}) \sum_{k=1}^{K} p_g(T_k|\tilde{x}) \log D_{\mathrm{LA}}(k, 0|\tilde{x}) d\tilde{x}. \tag{42}$$

Maximizing this integral is equivalent to maximize the component for every transformed data $\tilde{x} \in \tilde{\mathcal{X}}$:

$$\max_{D_{\mathrm{LA}}} p_d^T(\tilde{x}) \sum_{k=1}^{K} p_d(T_k|\tilde{x}) \log D_{\mathrm{LA}}(k, 1|\tilde{x}) + p_g^T(\tilde{x}) \sum_{k=1}^{K} p_g(T_k|\tilde{x}) \log D_{\mathrm{LA}}(k, 0|\tilde{x}), \tag{43}$$

$$\text{s.t.} \sum_{k=1}^{K} D_{\mathrm{LA}}(k, 1|\tilde{x}) + D_{\mathrm{LA}}(k, 0|\tilde{x}) = 1. \tag{44}$$

Define the Lagrange function as follows:

$$\mathcal{L}(D_{\mathrm{LA}}, \lambda) = \sum_{i=0}^{1} p_i^T(\tilde{x}) \sum_{k=1}^{K} p_i(T_k|\tilde{x}) \log D_{\mathrm{LA}}(k, i|\tilde{x}) + \lambda \left(\sum_{i=0}^{1} \sum_{k=1}^{K} D_{\mathrm{LA}}(k, i|\tilde{x}) - 1\right), \tag{45}$$

with $p_1^T(\tilde{x}) = p_d^T(\tilde{x})$, $p_0^T(\tilde{x}) = p_g^T(\tilde{x})$, $p_1(T_k|\tilde{x}) = p_d(T_k|\tilde{x})$ and $p_0(T_k|\tilde{x}) = p_g(T_k|\tilde{x})$, and $\lambda \in \mathbb{R}$ the Lagrange multiplier.

Calculate the derivatives with respect to $D_{\mathrm{LA}}(k, i|\tilde{x})$ and $\lambda$ and let them equals to 0, then we have:

$$\frac{\partial \mathcal{L}}{\partial D_{\mathrm{LA}}(k, i|\tilde{x})} = \frac{p_i^T(\tilde{x}) p_i(T_k|\tilde{x})}{D_{\mathrm{LA}}(k, i|\tilde{x})} + \lambda = 0, \frac{\partial \mathcal{L}}{\partial \lambda} = \sum_{i=0}^{1} \sum_{k=1}^{K} D_{\mathrm{LA}}(k, i|\tilde{x}) - 1 = 0. \tag{46}$$

By solving the above equations and according to $\frac{\partial^2 \mathcal{L}}{\partial D_{\mathrm{LA}}(k,i|\tilde{x})^2} = -\frac{p_i^T(\tilde{x})p_i(T_k|\tilde{x})}{(D_{\mathrm{LA}}(k,i|\tilde{x}))^2} < 0$, the optimal label-augmented discriminator for a transformed data $\tilde{x} \in \tilde{\mathcal{X}}$ can be expressed as follows:

$$D_{\mathrm{LA}}^*(k,i|\tilde{x}) = \frac{p_i^T(\tilde{x})p_i(T_k|\tilde{x})}{\sum_{i=0}^1 \sum_{k=1}^K p_i^T(\tilde{x})p_i(T_k|\tilde{x})} = \frac{p(T_k)p_i(\tilde{x}|T_k)}{\sum_{i=0}^1 \sum_{k=1}^K p(T_k)p_i(\tilde{x}|T_k)} \tag{47}$$

$$= \frac{p_i(\tilde{x}|T_k)}{\sum_{i=0}^1 \sum_{k=1}^K p_i(\tilde{x}|T_k)} = \frac{p_i(\tilde{x}|T_k)}{\sum_{k=1}^K p_0(\tilde{x}|T_k) + p_1(\tilde{x}|T_k)}. \tag{48}$$

The third equation holds because of $p(T_k) = \frac{1}{K}, \forall T_k \in \mathcal{T}$. This concludes the proof because of the defined notations: $p_1(\tilde{x}|T_k) = p_d(\tilde{x}|T_k) = p_d^{T_k}(\tilde{x})$ and $p_0(\tilde{x}|T_k) = p_g(\tilde{x}|T_k) = p_g^{T_k}(\tilde{x})$.

$\square$

## A.4  Proof of Theorem 3

**Theorem 3.** *The objective function for the generator of SSGAN-LA, given the optimal label-augmented discriminator, boils down to:*

$$\min_G \frac{1}{K} \sum_{k=1}^K \mathbb{D}_{\mathrm{KL}}(\mathcal{P}_g^{T_k} \| \mathcal{P}_d^{T_k}). \tag{11}$$

*The global minimum is achieved if and only if $\mathcal{P}_g = \mathcal{P}_d$ when $\exists T_k \in \mathcal{T}$ is an invertible transformation.*

*Proof.*

$$\max_G \mathbb{E}_{x \sim \mathcal{P}_g, T_k \sim \mathcal{T}}[\log D_{\mathrm{LA}}^*(k,1|T_k(x))] - \mathbb{E}_{x \sim \mathcal{P}_g} \mathbb{E}_{T_k \sim \mathcal{T}}[\log D_{\mathrm{LA}}^*(k,0|T_k(x))] \tag{49}$$

$$\Rightarrow \max_G \mathbb{E}_{\tilde{x} \sim \mathcal{P}_g^T, T_k \sim \mathcal{T}_g^{\tilde{x}}}[\log D_{\mathrm{LA}}^*(k,1|\tilde{x})] - \mathbb{E}_{\tilde{x} \sim \mathcal{P}_g^T, T_k \sim \mathcal{T}_g^{\tilde{x}}}[\log D_{\mathrm{LA}}^*(k,0|\tilde{x})] \tag{50}$$

$$\Rightarrow \max_G \mathbb{E}_{\tilde{x} \sim \mathcal{P}_g^T, T_k \sim \mathcal{T}_g^{\tilde{x}}} \left[ \log \frac{D_{\mathrm{LA}}^*(k,1|\tilde{x})}{D_{\mathrm{LA}}^*(k,0|\tilde{x})} \right] \tag{51}$$

$$\Rightarrow \max_G \int p_g^T(\tilde{x}) \sum_{k=1}^K p_g(T_k|\tilde{x}) \log \left( \frac{p_d(\tilde{x}|T_k)}{p_g(\tilde{x}|T_k)} \right) d\tilde{x} \tag{52}$$

$$\Rightarrow \max_G \sum_{k=1}^K p(T_k) \int p_g(\tilde{x}|T_k) \log \left( \frac{p_d(\tilde{x}|T_k)}{p_g(\tilde{x}|T_k)} \right) d\tilde{x} \tag{53}$$

$$\Rightarrow \min_G \frac{1}{K} \sum_{k=1}^K \mathbb{D}_{\mathrm{KL}}(\mathcal{P}_g^{T_k} \| \mathcal{P}_d^{T_k}). \tag{54}$$

Notice that $f$-divergence including the KL divergence is invariant to invertible/affine transformation [31, 47, 57, 59]. In other words, $\mathbb{D}_{\mathrm{KL}}(\mathcal{P}_g^{T_k} \| \mathcal{P}_d^{T_k}) = \mathbb{D}_{\mathrm{KL}}(\mathcal{P}_g \| \mathcal{P}_d)$ when the transformation $T_k$ is invertible, which induces $\mathcal{P}_g = \arg\min_G \mathbb{D}_{\mathrm{KL}}(\mathcal{P}_g^{T_k} \| \mathcal{P}_d^{T_k}) \# \mathcal{P}_z = \arg\min_G \mathbb{D}_{\mathrm{KL}}(\mathcal{P}_g \| \mathcal{P}_d) \# \mathcal{P}_z = \mathcal{P}_d$. In addition, $\mathcal{P}_g = \mathcal{P}_d$ is a sufficient condition for $\mathcal{P}_g^{T_k} = \mathcal{P}_d^{T_k}$ that fully minimizes $\mathbb{D}_{\mathrm{KL}}(\mathcal{P}_g^{T_k} \| \mathcal{P}_d^{T_k})$ regardless of the invertibility of $T_k$. Therefore, the global maximum of the objective function for the generator of SSGAN-LA under the optimal label-augmented discriminator is achieved if and only if $\mathcal{P}_g = \mathcal{P}_d$ when existing a transformation $T_k \in \mathcal{T}$ is invertible.

$\square$

## A.5  Proof of Theorem 4

**Theorem 4.** *At the equilibrium point of DAGAN, the optimal generator implies $\mathcal{P}_g^T = \mathcal{P}_d^T$. However, if $(\mathcal{T}, \circ)$ forms a group and $T_k \in \mathcal{T}$ is uniformly sampled, then the probability that the optimal generator replicates the real data distribution is $\mathbb{P}(\mathcal{P}_g = \mathcal{P}_d | \mathcal{P}_g^T = \mathcal{P}_d^T) = 0$.*

*Proof.* We first prove the first sentence in this Theorem. According to Proposition 2, the objective function of DAGAN can be rewritten as follows:

$$\min_G \max_D V(G, D) = \mathbb{E}_{x \sim \mathcal{P}_d, T_k \in \mathcal{T}}[\log D(T_k(x))] + \mathbb{E}_{x \sim \mathcal{P}_g, T_k \in \mathcal{T}}[\log(1 - D(T_k(x)))] \tag{55}$$

$$= \mathbb{E}_{\tilde{x} \sim \mathcal{P}_d^T, T_k \in \mathcal{T}_d^{\tilde{x}}}[\log D(\tilde{x})] + \mathbb{E}_{\tilde{x} \sim \mathcal{P}_g^T, T_k \in \mathcal{T}_g^{\tilde{x}}}[\log(1 - D(\tilde{x}))] \tag{56}$$

$$= \mathbb{E}_{\tilde{x} \sim \mathcal{P}_d^T}[\log D(\tilde{x})] + \mathbb{E}_{\tilde{x} \sim \mathcal{P}_g^T}[\log(1 - D(\tilde{x}))]. \tag{57}$$

According to the Theorem 1 in [14], the global minimum of the virtual training criterion $V(G, D^*)$, given the optimal discriminator $D^*$, is achieved if and only if $\mathcal{P}_g^T = \mathcal{P}_d^T$.

We then prove the second sentence in this Theorem. The main idea of the proof is to construct countless generated distributions who satisfy the equilibrium point of DAGAN. However, there is only one real data distribution. Therefore, the probability that the generator of DAGAN learns the real data distribution is $\frac{1}{\infty} = 0$ even though at its equilibrium point.

Since the set of transformations $\mathcal{T}$ forms a group with respect to the composition operator $\circ$, and according to the **closure property** of group, the composition of any two transformations is also in the set (i.e., $T_i \circ T_j \in \mathcal{T}, \forall T_i, T_j \in \mathcal{T}$). In addition, according to the converse-negative proposition of the **cancellation law** of group, the compositions of a transformation with other different transformations are different from each other (i.e., $T_i \circ T_k \neq T_j \circ T_k, \forall T_i \neq T_j, T_k \in \mathcal{T}$). Based on the above properties and **inclusion–exclusion principle**, we have $\{T_j \circ T_i | T_i \in \mathcal{T}\} = \mathcal{T}, \forall T_j \in \mathcal{T}$.

Let us construct a family of distribution $\mathcal{P}_\pi$ with density of $p_\pi(\hat{x}) = \sum_{j=1}^K \pi_j p_d(\hat{x}|T_j) = \sum_{j=1}^K \pi_j \int p_d(x) p(\hat{x}|x, T_j) dx$ with mixture weights $\Pi = \{\pi_j\}_{j=1}^K$, subject to $\sum_{j=1}^K \pi_j = 1$ and $0 \leq \pi_j \leq 1, \forall \pi_j \in \Pi$, then the mixture transformed distribution of data from $\mathcal{P}_\pi$ is:

$$p_\pi^T(\tilde{x}) = \int p_\pi(\hat{x}) \sum_{i=1}^K p(T_i) p(\tilde{x}|\hat{x}, T_i) d\hat{x} \tag{58}$$

$$= \int \sum_{j=1}^K \pi_j \int p_d(x) p(\hat{x}|x, T_j) dx \sum_{i=1}^K p(T_i) p(\tilde{x}|\hat{x}, T_i) d\hat{x} \tag{59}$$

$$= \sum_{j=1}^K \pi_j \int \sum_{i=1}^K p_d(x) p(\hat{x}|x, T_j) p(T_i) p(\tilde{x}|\hat{x}, T_i) dx d\hat{x} \tag{60}$$

$$= \sum_{j=1}^K \pi_j \int \sum_{i=1}^K p(T_i) p_d(x) p(\tilde{x}, \hat{x}|x, T_i, T_j) dx d\hat{x} \tag{61}$$

$$= \sum_{j=1}^K \pi_j \int \sum_{i=1}^K p(T_i) p_d(x) p(\tilde{x}|x, T_i, T_j) dx \tag{62}$$

$$= \sum_{j=1}^K \pi_j \int \sum_{i=1}^K \frac{1}{K} p_d(x) p(\tilde{x}|x, T_i, T_j) dx \tag{63}$$

$$= \sum_{j=1}^K \pi_j \int \sum_{k=1}^K \frac{1}{K} p_d(x) p(\tilde{x}|x, T_k) dx \tag{64}$$

$$= \sum_{j=1}^K \pi_j \int \sum_{k=1}^K p(T_k) p_d(x) p(\tilde{x}|x, T_k) dx \tag{65}$$

$$= \sum_{j=1}^K \pi_j p_d^T(\tilde{x}) \tag{66}$$

$$= p_d^T(\tilde{x}). \tag{67}$$

Equation 61 holds because of $p(\hat{x}|x, T_j) = p(\hat{x}|x, T_i, T_j)$ ($\hat{x}$ only depends on $x$ and $T_j$) and $p(\tilde{x}|\hat{x}, T_i) = p(\tilde{x}|\hat{x}, x, T_i, T_j)$ ($\tilde{x}$ is independent of $x$ and $T_j$ given $\hat{x}$ and $T_i$). Equation 64 holds because of $p(\tilde{x}|x, T_i, T_j) = \delta(\tilde{x} - T_j \circ T_i(x))$ and $\{T_j \circ T_i | T_i \in \mathcal{T}\} = \mathcal{T}, \forall T_j \in \mathcal{T}$.

Therefore, there are infinite generator distributions ($\mathcal{P}_\pi$) that satisfy the equilibrium point of DAGAN ($\mathcal{P}_\pi^T = \mathcal{P}_d^T$), but only one of them is the target distribution (i.e., the real data distribution $\mathcal{P}_d$). In particular, we have $\mathcal{P}_\pi = \mathcal{P}_d$ if and only if $\pi_1 = 1, \pi_k = 0, \forall k = \{2, 3, \cdots, K\}$, and the corresponding probability is $\mathbb{P}(\mathcal{P}_\pi = \mathcal{P}_d | \mathcal{P}_\pi^T = \mathcal{P}_d^T) = \mathbb{P}(\pi_1 = 1, \pi_k = 0, \forall \pi_k \in \Pi \setminus \pi_1 | \sum_{k=1}^{K} \pi_k = 1, 0 \leq \pi_k \leq 1, \forall \pi_k \in \Pi) = 0$. This concludes our proof. $\qquad\square$

## B Experimental Settings in Section 5.3

We implement all methods based on unconditional BigGAN [4] without utilizing the annotated class labels. To construct the unconditional BigGAN, we replace the conditional batch normalization [9] in the generator with standard batch normalization [20] and remove the label projection technique [39] in the discriminator. The network architecture of generators of all methods is the same, and the network architecture of discriminators of all methods is different only in the output layer. We train all methods for 100 epochs with a batch size of 100 on all datasets. The optimizer is Adam with betas $(\beta_1, \beta_2) = (0.0, 0.999)$ for both the generator and discriminator. The learning rate for the generator is $2 \times 10^{-4}$ on CIFAR-10 and STL-10, and $1 \times 10^{-4}$ on Tiny-ImageNet, and the learning rate for the discriminator/classifier is $2 \times 10^{-4}$ on CIFAR-10 and STL-10, and $4 \times 10^{-4}$ on Tiny-ImageNet. All baselines use the hinge loss [32, 56] as the implementation of the original GAN loss.

$$\min_D \mathcal{L}_D^H = \mathbb{E}_{x \sim \mathcal{P}_d}[\max(0, 1 - D(x))] + \mathbb{E}_{x \sim \mathcal{P}_g}[\max(0, 1 + D(x))], \tag{68}$$

$$\min_G \mathcal{L}_G^H = \mathbb{E}_{x \sim \mathcal{P}_g}[-D(x)], \tag{69}$$

where the output range of the discriminator $D : \mathcal{X} \to \mathbb{R}$ is unconstrained.

Analogously, we use the multi-class hinge loss to implement the objective functions of SSGAN-LA:

$$\min_{D_{LA}} \mathcal{L}_{D_{LA}}^{MH} = \mathbb{E}_{x \sim \mathcal{P}_d, T_k \sim \mathcal{T}}[\mathbb{E}_{(t,l) \neq (k,1)}[\max(0, 1 - D_{LA}(k, 1|T_k(x)) + D_{LA}(t, l|T_k(x))))]]$$

$$+ \mathbb{E}_{x \sim \mathcal{P}_g, T_k \sim \mathcal{T}}[\mathbb{E}_{(t,l) \neq (k,0)}[\max(0, 1 - D_{LA}(k, 0|T_k(x)) + D_{LA}(t, l|T_k(x))))]], \tag{70}$$

$$\min_G \mathcal{L}_G^{MH} = \mathbb{E}_{x \sim \mathcal{P}_g, T_k \sim \mathcal{T}}[\mathbb{E}_{(t,l) \neq (k,1)}[-D_{LA}(k, 1|T_k(x)) + D_{LA}(t, l|T_k(x)))]]$$

$$- \mathbb{E}_{x \sim \mathcal{P}_g, T_k \sim \mathcal{T}}[\mathbb{E}_{(t,l) \neq (k,0)}[-D_{LA}(k, 0|T_k(x)) + D_{LA}(t, l|T_k(x)))]], \tag{71}$$

where the label-augmented discriminator $D_{LA} : \tilde{\mathcal{X}} \to \mathbb{R}^{2K}$ is no longer required to output a softmax probability. The multi-class hinge loss functions are the extended version of the (binary) hinge loss functions. Other hyper-parameters in baselines are the same as the authors' suggestions in their papers unless otherwise specified. To obtain the optimal label-augmented discriminator of SSGAN-LA as much as possible and because that the discriminator solves a more challenging classification task, we set the discriminator updating steps per generator step as $n_{dis} = 4$ for SSGAN-LA. We follow the practices in [13, 28, 59] to perform all transformations on each sample for DAGAN, DAGAN-MD, and SSGAN-LA.

## C Performance of SSGAN-LA with the Original Discriminator

We investigate the performance of the proposed label-augmented discriminator $D_{LA}$ combined with the original discriminator $D$ under the same data transformation setting as SSGAN [6] that rotate a quarter images in a batch in all four considered directions. Specifically, the objective functions for the original discriminator $D$, the label-augmented discriminator $D_{LA}$, and the generator $G$ of the original discriminator retained SSGAN-LA (SSGAN-LA$^+$) are formulated as the following:

$$\min_{D, D_{LA}} \mathcal{L}_D^H + \lambda_d \cdot \mathcal{L}_{D_{LA}}^{MH}, \tag{72}$$

$$\min_G \mathcal{L}_G^H + \lambda_g \cdot \mathcal{L}_G^{MH}, \tag{73}$$

where $\lambda_d$ and $\lambda_g$ are two hyper-parameters that trade-off the GAN task and the self-supervised task. The discriminator update steps are 2 per generator update step for all methods following the practice of SSGAN. The hyper-parameters $\lambda_d$ and $\lambda_g$ of SSGAN, SSGAN-MS, and SSGAN-LA$^+$ are selected from $\{0.2, 1.0\}$ according to the best FID result. As reported in Table 4, SSGAN-LA$^+$ outperforms all competitive baselines in terms of both FID and IS results, verifying the effectiveness of the proposed self-supervised approach for training GANs.

Table 4: FID ($\downarrow$) and IS ($\uparrow$) comparison on CIFAR-10, STL-10, and Tiny-ImageNet. SSGAN-LA$^+$ retains the original discriminator. We use the same data transformation setting of SSGAN.

| Dataset | Metric | GAN | SSGAN | SSGAN-MS | DAGAN$^+$ | DAGAN-MD | SSGAN-LA$^+$ |
|---------|--------|-----|-------|----------|-----------|----------|--------------|
| CIFAR-10 | FID | 10.83 | 7.52 | 7.08 | 11.07 | 10.05 | **6.64** |
| | IS | 8.42 | 8.29 | 8.45 | 8.10 | 8.14 | **8.51** |
| STL-10 | FID | 20.15 | 16.84 | 16.46 | 18.97 | 21.68 | **15.91** |
| | IS | 10.25 | 10.36 | 10.40 | 10.12 | 10.29 | **10.87** |
| Tiny-ImageNet | FID | 31.01 | 30.09 | 25.76 | 58.91 | 50.14 | **24.23** |
| | IS | 9.80 | 10.21 | 10.57 | 7.06 | 7.80 | **10.86** |

# D  Ablation Study on $\lambda_g$ of SSGAN and SSGAN-MS

In this experiment, we investigate the effects of the self-supervised task for the generator in SSGAN [6] and SSGAN-MS [57]. We change the value of $\lambda_g$ while keeping $\lambda_d = 1.0$ fixed. In particular, $\lambda_g = 0.0$ means that the generator is trained without self-supervised tasks. As reported in Table 5, SSGAN and SSGAN-MS with the authors' suggested $\lambda_g$ (i.e., $\lambda_g^*$) generally perform better than those with $\lambda_g = 0.0$, respectively, showing that the self-supervised task for the generator could benefit the generation performance of the generator. Arguably, the reason is that the self-supervised task can reduce the difficulty of optimization of the generator by providing additional useful guidance. However, as $\lambda_g$ increases to 1.0, the generation performance reflected by the FID scores show degradation. This verifies that the implied learning objective in the self-supervised tasks for the generator of SSGAN and SSGAN-MS are essentially inconsistent with the task of generative modeling.

Table 5: Ablation study on the hyper-parameter $\lambda_g$ of SSGAN and SSGAN-MS.

| Dataset | Metric | SSGAN | | | SSGAN-MS | | |
|---------|--------|----------------|------------------------|----------------|----------------|------------------------|----------------|
| | | $\lambda_g = 0.0$ | $\lambda_g = \lambda_g^*$ | $\lambda_g = 1.0$ | $\lambda_g = 0.0$ | $\lambda_g = \lambda_g^*$ | $\lambda_g = 1.0$ |
| CIFAR-10 | FID | 8.07 | 7.52 | 8.54 | 7.08 | 7.16 | 6.81 |
| | IS | 8.21 | 8.29 | 8.41 | 8.45 | 8.45 | 8.26 |
| STL-10 | FID | 18.04 | 16.84 | 18.88 | 17.5 | 16.46 | 19.26 |
| | IS | 10.20 | 10.36 | 10.03 | 10.60 | 10.40 | 10.01 |
| Tiny-ImageNet | FID | 30.69 | 30.09 | 30.27 | 99.74 | 25.76 | 26.44 |
| | IS | 10.67 | 10.21 | 10.23 | 6.11 | 10.57 | 10.61 |