# OpenReview forum: "Self-Supervised GANs with Label Augmentation"
_NeurIPS.cc/2021/Conference — NeurIPS 2021 Poster_

### Official Review · Reviewer_VBjE · 2021-07-08

**Rating:** 6
**Confidence:** 3

**Summary:**

In the context of using self-supervision signals in the GAN training framework, this paper proposes to use a multi-class classification approach in the discriminator paired with the task of predicting the rotation of an image.  This is proposed instead of using two separate discriminators: one for the rotation prediction and the other for the real/fake classification.
The idea supposes a small and simple change to existing methods that, although in terms of novelty it is limited, it is justified theoretically and quantitative empirical results support the idea.

**Limitations And Societal Impact:**

The authors have not addressed the societal impact.

**Main Review:**

The paper is generally well written and clear. In my opinion, the experimental section is sufficient and it is tested on enough datasets. The idea is simple but well motivated, and it improves over the self-supervised baselines.

Although the paper states that it does not aim to compare against state of the art data augmentation techniques, it is a very important comparison to do. If data augmentation works better than self-supervision in GANs, why use self-supervision to begin with? In this case, it is not clear that the data augmentations can directly be combined with the self-supervised rotation signals.

Moreover, the section regarding data augmentations in the paper (and the baseline comparisons) revolve around rotations, which is not the best data augmentation strategy one could use. Moreover, the paper talks about transformation leaking, when in [1], experimental results show that, as long as the non-transformed (0 degrees) image is seen enough times, leakage should not occur. Why was this not considered in this paper, whether it is in the discussion part and/or the experimental results? Could the authors elaborate on this? My concern is that the data augmentation baselines might be at a disadvantage and therefore, be an unfair comparison.

As an additional thought, the idea of using rotations as self-supervision signals in the self-supervision community is arguably outdated. There is other self-supervision tasks or transformations to predict in the discriminator that could be borrowed from the self-supervised learning community to apply them to GAN training, and perhaps be more competitive with the extensive data augmentation proposed in [1], for example.

L.90: if I understand correctly, the generator wants to fool the rotation classifier as much as if wants to fool the discriminator, so
I suggest changing the term “collaborate” for another more appropriate word.

[1] Zhao, S., Liu, Z., Lin, J., Zhu, J. Y., & Han, S. (2020). Differentiable augmentation for data-efficient gan training. arXiv preprint arXiv:2006.10738.


**Time Spent Reviewing:**

2

---

> ### Author Response · Authors · 2021-08-10
> **Response to Reviewer VBjE**
>
> We would like to thank the reviewer for providing positive comments on this paper. We are encouraged that the reviewer thought that our approach is well motivated and adequately validated by enough experiments.
>
> ---
>
> **Q1:** Although the paper states that it does not aim to compare against state of the art data augmentation techniques, it is a very important comparison to do. If data augmentation works better than self-supervision in GANs, why use self-supervision to begin with? In this case, it is not clear that the data augmentations can directly be combined with the self-supervised rotation signals.
>
> **R1:** The proposed SSGAN-LA is an extension of data augmentation GANs (DAGANs). Specifically, the discriminator of DAGANs (including StyleGAN2-ADA and DiffAugment) takes the form of $D(i|T_k(x))$ while the discriminator of SSGAN-LA takes the form of $D(k,i|T_k(x))$. Compared to DAGANs, the discriminator of SSGAN-LA not only distinguishes real ($i=1$) from fake ($i=0$) but also recognizes the performed transformation signal $k$ on data $x$. Notice that rotations are also applied as data augmentations in the GAN literature [A] and the proposed SSGAN-LA is not limited to rotations. In other words, $T_k$ represents any kind of data augmentation. Therefore, it is natural to combine the data augmentation methods with self-supervised signals. For example, one can extend the data augmentation methods to the proposed self-supervised methods by predicting the performed augmentation of interest (e.g., rotations) in the discriminator as well. Furthermore, predicting the self-supervision signals would enable the discriminator to learn stable and better representations (as shown in the representation learning experiments in Section 5.2.2 and Appendix C.3) to provide more informative guidance to the generator. We report the comparison with StyleGAN-ADA in the following table to show the effectiveness of SSGAN-LA compared to SOTA data augmentation GAN. We train both methods for 20,000 kimg for a fair comparison and due to our limited computational resource.
>
> | Methods                                                      | FID  |
> | ------------------------------------------------------------ | ---- |
> | StyleGAN2-ADA [A]                                            | 3.49 |
> | SSGAN-LA (based on StyleGAN2-ADA by predicting the rotation signals as well in the discriminator) | 3.40 |
>
>
> ---
>
>
> **Q2:** Moreover, the section regarding data augmentations in the paper (and the baseline comparisons) revolve around rotations, which is not the best data augmentation strategy one could use. Moreover, the paper talks about transformation leaking, when in [1], experimental results show that, as long as the non-transformed (0 degrees) image is seen enough times, leakage should not occur. Why was this not considered in this paper, whether it is in the discussion part and/or the experimental results? Could the authors elaborate on this? My concern is that the data augmentation baselines might be at a disadvantage and therefore, be an unfair comparison.
>
> **R2:** Indeed, upweighting the probability that the non-transformed images would avoid the data leakage issue. However, we introduce DAGAN and view it as an ablation model of SSGAN-LA to show the importance of utilizing self-supervision. DAGAN-MD, which fixes this data leaking issue, is still inferior to SSGAN-LA in both generative modeling and representation learning, validating the importance of utilizing self-supervision.
>
> ---
>
> **Q3:** As an additional thought, the idea of using rotations as self-supervision signals in the self-supervision community is arguably outdated. There are other self-supervision tasks or transformations to predict in the discriminator that could be borrowed from the self-supervised learning community to apply them to GAN training, and perhaps be more competitive with the extensive data augmentation proposed in [1], for example.
>
> **R3:** We follow the practice of existing self-supervised GANs (SSGANs) to use rotations as the transformation as we aim to solve the problem of existing SSGANs. Using rotations as self-supervision signals is active as recent studies also focus on rotations [A, B]. We would like to thank the reviewer for providing valuable suggestions and will explore more self-supervision signals in future work.
>
> ---
>
> **Q4:** L.90: if I understand correctly, the generator wants to fool the rotation classifier as much as if it wants to fool the discriminator, so I suggest changing the term “collaborate” for another more appropriate word.
>
> **R4:** In fact, the generator does not aim to fool the rotation classifier. Specifically, we have $\min_G - \mathbb{E}[\log C(k|T_k(x))] \Rightarrow \max_G \mathbb{E}[\log C(k|T_k(x))], x \sim G, T_k \sim \mathcal{T}$, which means that the generator is collaborate with the classifier on the self-supervised task ($\max_C \mathbb{E}[\log C(k|T_k(x))]$).
>
>
>
> [A] Karras, Tero, et al. "Training generative adversarial networks with limited data." *arXiv preprint arXiv:2006.06676* (2020).
>
> [B] Reed, Colorado J., et al. "Selfaugment: Automatic augmentation policies for self-supervised learning." *Proceedings of the IEEE/CVF Conference on Computer Vision and Pattern Recognition*. 2021.

---

> > ### Author Response · Authors · 2021-08-30
> > **Comparison with SOTA data augmentation GANs**
> >
> > Dear Reviewer VBjE,
> >
> > We would like to thank you once again for your constructive feedback. In response to your concern on comparison with data augmentation methods, we have added the experiments of our method comparing ADA [a] and DiffAugment [b], respectively, under their settings. Experimental results are reported as follows.
> >
> > | FID                         | CIFAR-10 |
> > | --------------------------- | -------- |
> > | StyleGAN2-ADA [a]           | 2.92     |
> > | + Label Augmentation (ours) | **2.68** |
> >
> > | FID (10k)                   | CIFAR-10 | CIFAR-100 |
> > | --------------------------- | -------- | --------- |
> > | BigGAN + DiffAugment [b]    | 8.70     | 12.00     |
> > | + Label Augmentation (ours) | **8.05** | **9.77**  |
> >
> > The above results demonstrate that our proposed method significantly outperforms existing state-of-the-art data augmentation GANs (ADA [a] and DiffAugment [b]), showing that the label augmentation method we proposed is a complement to the existing data augmentation method. We will report the results in the updated version.
> >
> > Best regards,
> >
> > The authors
> >
> > [a] Karras, Tero, et al. "Training Generative Adversarial Networks with Limited Data." *Advances in Neural Information Processing Systems* 33 (2020).
> >
> > [b] Zhao, Shengyu, et al. "Differentiable Augmentation for Data-Efficient GAN Training." *Advances in Neural Information Processing Systems* 33 (2020).

---

### Official Review · Reviewer_qAsa · 2021-07-14

**Rating:** 5
**Confidence:** 5

**Summary:**

The paper presents an approach for training GANs with transformation-based self-supervision. It unifies the original GAN task with the self-supervised task into a single multi-class classification task for the discriminator by augmenting the real/fake labels with the self-supervised labels. The authors demonstrate that the proposed approach outperforms some existing transformation-based self-supervised and data augmentation GANs using experiments on CIFAR-10, STL-10 and Tiny-ImageNet. They also provide a theoretical analysis of their method.

**Limitations And Societal Impact:**

Yes

**Main Review:**

**Strengths**:


- The proposed approach is intuitive, and outperforms some of the existing self-supervised and augmentation GANs.


**Weaknesses**:


- There are other works not included in comparisons. Karras et al. [A] present a method for training GANs with limited data, which achieves an FID of 2.92 on CIFAR-10. This is better than the FID of 5.87 reported in this paper. Another relevant work is [53] (differentiable augmentation GAN) which is not included in comparisons. The authors mention that "we do not aim to compare with state-of-the-art data augmentation GANs in this work" (L 179). This is unclear to me as they already provide a comparison with another augmentation method (DAGAN). Both self-supervised and augmentation GAN approaches aim to improve the final generation quality, so comparison with methods in both of these domains is expected.


-  Contributions of the paper are modest for this venue. The proposed approach is quite similar to SSGAN(-MS), with some relatively minor differences such as applying self-supervision to real samples as well. Theoretical analysis in the paper is also similar to the original GAN paper.


- Resolution and complexity of the datasets used in the paper are limited. The maximum resolution used is 64x64 for Tiny ImageNet. [A, 53] show results on higher resolution datasets.


[A] Training Generative Adversarial Networks with Limited Data; Karras et al.; NeurIPS 2020

**Time Spent Reviewing:**

8

---

> ### Author Response · Authors · 2021-08-10
> **Response to Reviewer qAsa**
>
> We would like to thank the reviewer for providing valuable feedback on this paper. To highlight the contribution of our work, we address each of the weaknesses pointed out by the reviewer.
>
> ---
>
> **W1:** There are other works not included in comparisons. Karras et al. [A] present a method for training GANs with limited data, which achieves an FID of 2.92 on CIFAR-10. This is better than the FID of 5.87 reported in this paper. Another relevant work is [53] (differentiable augmentation GAN) which is not included in comparisons.
>
> **R1:** We argue that the proposed SSGAN-LA is complementary to existing data augmentation GANs.
>
> StyleGAN2-ADA [A] achieves an FID of 2.92 on CIFAR-10 based on StyleGAN2 instead of BigGAN that the proposed SSGAN-LA is based on. Modifying network architecture would significantly affect experimental results. In addition, StyleGAN2-ADA is trained much longer than ours (2000 epochs v.s. 100 epochs).
>
> As we mentioned in line 177, DiffAugment [53] is mathematically equivalent to DAGAN, which we have compared in this work. Specifically, the discriminator of DAGANs (e.g., StyleGAN2-ADA and DiffAugment) takes the form of $D(i|T_k(x))$ while SSGAN-LA takes the form of $D(k,i|T_k(x))$. Compared to DAGAN, the discriminator of SSGAN-LA not only distinguishes real ($i=1$) from fake ($i=0$) but also recognizes the performed transformation $T_k$ on data $x$. In other words, we view DAGAN as an ablation model of SSGAN-LA. Extensive experiments show that recognizing the self-supervision (performed transformation) as well as realism benefits for generation quality. Therefore we argue that SSGAN-LA is able to outperform StyleGAN2-ADA and DiffAugment if they are in a fairness data augmentation setting and will conduct this comparison in the updated version.
>
> We report the comparison with StyleGAN-ADA in the followint table to show the effectiveness of SSGAN-LA compared to SOTA data augmentation GAN. We train both methods for 20,000 kimg for a fair comparison and due to our limited computational resource.
>
> | Methods                                                      | FID  |
> | ------------------------------------------------------------ | ---- |
> | StyleGAN2-ADA [A]                                            | 3.49 |
> | SSGAN-LA (based on StyleGAN2-ADA by predicting the rotation signals as well in the discriminator) | 3.40 |
>
> ---
>
> **W2:** Contributions of the paper are modest for this venue. The proposed approach is quite similar to SSGAN(-MS), with some relatively minor differences such as applying self-supervision to real samples as well. Theoretical analysis in the paper is also similar to the original GAN paper.
>
> **R2:** In fact, the proposed SSGAN-LA applies self-supervision to fake samples relative to SSGAN(-MS). Although being similar to SSGAN(-MS), the proposed SSGAN-LA solves the essential problem of SSGAN(-MS), which causes an inconsistent goal with the generative modeling, motivated by our novel understanding of this inconsistent goal.
>
> Our theoretical analysis follows the common practice in the GAN literature, and clearly shows the novelty and rationality of our method compared to existing self-supervised GANs. Both theoretical and experimental results show that SSGAN-LA have significant advantages over SSGAN-MS, which was published in this venue.
>
> In addition, we found that the original theoretical analysis of SSGAN-MS in their paper is wrong and derived the correct version (Theorem 2). In particular, the SSGAN-MS's authors mistakenly regard $\frac{p_d^T(y)}{p_g^T(y)}=\frac{\sum_k p_d(y|T_k)p(T_k)}{\sum_k p_g(y|T_k)p(T_k)}$ as $\frac{p_d(y|T_k)}{p_g(y|T_k)}$ in their proof. For more details, please compare our proof (in Appendix A.2) with theirs (in Appendix of SSGAN-MS [B]).
>
> Considering the proposed method is well-motivated and theoretically guaranteed, we argue that the contributions of this paper are sufficient for this venue.
>
> ---
>
> **W3:** Resolution and complexity of the datasets used in the paper are limited. The maximum resolution used is 64x64 for Tiny ImageNet. [A, 53] show results on higher resolution datasets.
>
> **R3:** Due to our limited computational resources, i.e., 2x NVIDIA Tesla K80 GPUs (4 FLOPS), running experiments on high resolution dataset in [A, 53], e.g., ImageNet, is difficult for us. According to the BigGAN-PyTorch repository, running experiments on ImageNet on 8x V100 GPUs (14 TFLOPS) will take 15 days. Therefore, running it on 2x K80 GPUs is expected to require $(14 * 8 * 15) / (4 * 2) / 30 = 7$ months, which is unaffordable for us. To our best efforts, we conducted experiments on a subset of ImageNet, i.e., Tiny-ImageNet (shares similar properties with ImageNet), to show the effectiveness of the proposed SSGAN-LA.
>
> ---
>
> [A] Training Generative Adversarial Networks with Limited Data; Karras et al.; NeurIPS 2020
>
> [B] Tran, Ngoc-Trung, et al. "Self-supervised GAN: Analysis and Improvement with Multi-class Minimax Game." *Advances in Neural Information Processing Systems* 32 (2019): 13253-13264.

---

> > ### Author Response · Authors · 2021-08-30
> > **Comparison with SOTA data augmentation GANs**
> >
> > Dear Reviewer qAsa,
> >
> > We would like to thank you once again for your constructive feedback. In response to your first question, we have added the experiments of our method comparing ADA [a] and DiffAugment [b], respectively, under their settings. Experimental results are reported as follows.
> >
> > | FID                         | CIFAR-10 |
> > | --------------------------- | -------- |
> > | StyleGAN2-ADA [a]           | 2.92     |
> > | + Label Augmentation (ours) | **2.68** |
> >
> > | FID (10k)                   | CIFAR-10 | CIFAR-100 |
> > | --------------------------- | -------- | --------- |
> > | BigGAN + DiffAugment [b]    | 8.70     | 12.00     |
> > | + Label Augmentation (ours) | **8.05** | **9.77**  |
> >
> > The above results demonstrate that our proposed method significantly outperforms existing state-of-the-art data augmentation GANs (ADA [a] and DiffAugment [b]), showing that the label augmentation method we proposed is a complement to the existing data augmentation method. We will report the results in the updated version.
> >
> > Best regards,
> >
> > The authors
> >
> > [a] Karras, Tero, et al. "Training Generative Adversarial Networks with Limited Data." *Advances in Neural Information Processing Systems* 33 (2020).
> >
> > [b] Zhao, Shengyu, et al. "Differentiable Augmentation for Data-Efficient GAN Training." *Advances in Neural Information Processing Systems* 33 (2020).

---

> > > ### Comment · Reviewer_qAsa · 2021-08-31
> > > **After Rebuttal**
> > >
> > > I thank the authors for the rebuttal. The new experiments show improved performance when combined with augmentation methods. However, I am still not convinced about adequate novelty of the proposed approach. The limited resolution of the datasets is another weakness of the paper, and lack of resources is not a proper justification. Finally, the theoretical analysis does not add major contributions compared to the original GAN paper. Therefore, I keep my rating.

---

> > > > ### Author Response · Authors · 2021-09-01
> > > > **Further clarification**
> > > >
> > > > We thank the reviewer for the further feedback.
> > > >
> > > >
> > > >
> > > > **Resolution of datasets**
> > > >
> > > > R: SSGAN-MS [a], which was published at NeurIPS 2019, conducted experiments with the highest resolution of 64x64 (CelebA), which equals the Tiny-ImageNet we experimented with. Recent studies [b,c,d,e] also conducted experiments on datasets with resolution equal to or lower than 64x64. We will conduct experiments on datasets with higher resolution in the updated version under our affordable settings. All the experimental results in the current paper show that our method significantly outperforms competing methods including SSGAN and SSGAN-MS.
> > > >
> > > >
> > > >
> > > > **Contribution of theoretical analysis**
> > > >
> > > > R: Our theoretical analysis is designed to show that the proposed SSGAN-LA can faithfully learn the real data distribution compared to existing self-supervised GANs (SSGAN and SSGAN-MS) but not the original GAN. The original GAN paper shows that its generator can faithfully learn the real data distribution. However, existing self-supervised GANs (SSGAN and SSGAN-MS) destroy the theoretical guarantee of the original GAN due to the introduction of inappropriate self-supervised tasks. Our proposed SSGAN-LA solves this problem of existing self-supervised GANs by introducing an appropriate self-supervised task that can also alleviate the discriminator catastrophic forgetting issue of the  original GAN. In summary, our method is an unbiased self-supervised GAN that can replicate the real data distribution by inheriting the convergence point of the original GAN.
> > > >
> > > > In addition, our theorem 2 corrects the error in the SSGAN-MS paper [a] and our theorem 4 shows the importance of utilizing self-supervised signals.
> > > >
> > > > [a] Tran, Ngoc-Trung, et al. "Self-supervised GAN: Analysis and Improvement with Multi-class Minimax Game." *Advances in Neural Information Processing Systems* 32 (2019): 13253-13264.
> > > >
> > > > [b] Sun, Ruoyu, Tiantian Fang, and Alexander Schwing. "Towards a better global loss landscape of GANs." *Advances in Neural Information Processing Systems* 33 (2020).
> > > >
> > > > [c] Ding, Xin, et al. "CcGAN: Continuous Conditional Generative Adversarial Networks for Image Generation." *International Conference on Learning Representations*. 2020.
> > > >
> > > > [d] Chavdarova, Tatjana, et al. "Taming GANs with Lookahead-Minmax." *International Conference on Learning Representations*. 2020.
> > > >
> > > > [e] Watanabe, Tomoki, and Paolo Favaro. "A Unified Generative Adversarial Network Training via Self-Labeling and Self-Attention." *International Conference on Machine Learning*. 2021.

---

> > > > ### Author Response · Authors · 2021-09-03
> > > > **Our method achieves new state-of-the-art performance on limited data**
> > > >
> > > > Dear Reviewer,
> > > >
> > > > Our new experiments show that the proposed label augmentation method is able to outperform competing methods and achieves new state-of-the-art results under limited data. As SSGAN-lA is complementary to data augmentation methods such DiffAugment [a], we follow their experimental settings to conduct experiments under limited data. Experimental results are shown in the following table, showing that our proposed self-supervised method has strong superiority in data efficiency compared to existing methods, including data augmentation (DiffAugment) and regularization (CR-BigGAN [b] and LeCam-GAN [c]).
> > > >
> > > > | FID                                          | CIFAR-10 100% | CIFAR-10 20% | CIFAR-10 10% | CIFAR-100 100% | CIFAR-100 20% | CIFAR-100 10% |
> > > > | -------------------------------------------- | ------------- | ------------ | ------------ | -------------- | ------------- | ------------- |
> > > > | BigGAN + DiffAugment (DA)                    | 8.70          | 14.04        | 22.40        | 12.00          | 22.14         | 33.70         |
> > > > | CR-BigGAN + DA                               | 8.49          | 12.84        | 18.70        | 11.25          | 20.28         | 26.90         |
> > > > | BigGAN + DA + LA (**ours**)                  | **8.05**      | **11.16**    | **15.08**    | **9.77**       | **15.91**        | **23.17**     |
> > > > | BigGAN + DA + Regularization (LeCam-GAN [c]) | 8.46          | 12.56        | 16.69        | 11.20          | 18.03         | 27.63         |
> > > >
> > > > In order to highlight the effectiveness of the proposed method, we will report the results in the updated version.
> > > >
> > > > Sincerely,
> > > >
> > > > The authors
> > > >
> > > > [a] Zhao, Shengyu, et al. "Differentiable Augmentation for Data-Efficient GAN Training." *Advances in Neural Information Processing Systems* 33 (2020).
> > > >
> > > > [b] Zhang, Han, et al. "Consistency Regularization for Generative Adversarial Networks." International Conference on Learning Representations. 2019.
> > > >
> > > > [c] Tseng, Hung-Yu, et al. "Regularizing Generative Adversarial Networks under Limited Data." *Proceedings of the IEEE/CVF Conference on Computer Vision and Pattern Recognition*. 2021.

---

> > > > > ### Public Comment · ~Lu_Jiang1 · 2022-02-12
> > > > > **Reported results in the updated version?**
> > > > >
> > > > > I wonder where the reported results in the updated version?
> > > > > I checked the latest version but could not find: https://arxiv.org/abs/2106.08601

---

### Official Review · Reviewer_f1xR · 2021-07-15

**Rating:** 7
**Confidence:** 4

**Summary:**

The authors propose a label augmentation GAN that combines the discriminator and the classifier in self-supervised learning GANs and show the priority on a variety of generative modeling tasks.

**Limitations And Societal Impact:**

The authors do not mention negative societal impact. One negative impact can be some improper use of generative models, such as DeepFake.

**Main Review:**

originality: The idea is novel. Combination of the classifier and the discriminator sounds natural and is an improvement over self-supervised GANs.

quality: The theorems look correct and the experiments seems solid. The only thing is that the numbers reported by the authors in Table seems to be different from BigGAN and SSGAN-MS. For example, BigGAN authors reportThe author may need to explain why. For example, the BigGAN authors report IS 9.22 and FID 14.73 which are different from the authors' IS 8.42 and FID 10.83. It may be an architecture difference but why not use the same architecture as BigGAN.

clarity: The paper is clearly written. The authors may cite [45] for theorem 2 and change "discriminator" to "classifier" in line 108. It seems that "classifier" and "generator" are competing against each other in the context.

significance: The only concern I have about the paper is the significance. The method seems to be novel but not significant. It seems to be a natural extension of self-supervised GANs. The paper seems to contain some theorems but there are still many loose points. It's unclear to me why self-supervised GAN can learn "stable" and "high-level" representations. I may be careless and skip some important details. Could the authors help me find which theorem in the paper gives the guarantee that the representation learned is "stable" and "high-level"?

**Time Spent Reviewing:**

5

---

> ### Author Response · Authors · 2021-08-10
> **Response to Reviewer f1xR**
>
> We would like to thank the reviewer for providing positive comments on this paper. We are encouraged that the reviewer thought that our idea is novel, our theorems are correct, and our experiments are solid.
>
>
>
> **Quality**
>
> **Q1:** The only thing is that the numbers reported by the authors in Table seem to be different from BigGAN and SSGAN-MS. For example, the BigGAN authors report IS 9.22 and FID 14.73 which are different from the authors' IS 8.42 and FID 10.83. It may be an architecture difference but why not use the same architecture as BigGAN.
>
> **R1:** In this work, we train all methods without annotated labels as we are studying unsupervised GAN. In other words, the BigGAN we compared is trained unsupervisedly, which is different from the original BigGAN that is trained supervisedly. To do that, we ignore the labels of the training data, and replace the conditional batch normalization (cBN) in the generator with BN and remove the label projection technique in the discriminator. Therefore, we cannot use the same architecture as the original (conditional) BigGAN. As for SSGAN-MS, they implemented SSGAN-MS based on SN-GAN or ResNet architecture, which is arguably outdated. In our experiments, we implemented all methods based on (unconditional) BigGAN for fairness.
>
>
>
> **Clarity**
>
> **Q2:** The authors may cite [45] for theorem 2 and change "discriminator" to "classifier" in line 108. It seems that "classifier" and "generator" are competing against each other in the context.
>
> **R2:** Actually, we deliberately did not cite [45] for Theorem 2 because it is different from the original one in [45]. They are different in the first term. The first term of Theorem 2 in [45] is $\frac{1}{K}[\sum_{k=1}^K \text{KL}(P_g^{T_k}\|P_d^{T_k})]$ while ours is $\text{KL}(P_g^{T}\|P_d^{T})$. We argue that ours is the correct version, and the authors of [45] get the wrong version by mistakely regarding  $\frac{p_d^T(y)}{p_g^T(y)}=\frac{\sum_k p_d(y|T_k)p(T_k)}{\sum_k p_g(y|T_k)p(T_k)}$ as $\frac{p_d(y|T_k)}{p_g(y|T_k)}$ in their proof. We view this new theorem as one of our contributions and thus did not cite [45]. For more details, please compare our proof (in Appendix A.2) with theirs (in Appendix of [45]), respectively. As for the typo in line 108, we would like to thank the reviewer and will fix it in the updated version.
>
>
>
> **Significance**
>
> **Q3:** It's unclear to me why self-supervised GAN can learn "stable" and "high-level" representations. Could the authors help me find which theorem in the paper gives the guarantee that the representation learned is "stable" and "high-level"?
>
> **R3:** Self-supervised GANs (SSGANs) can learn stable representations due to the fact that the learning environment of the self-supervised task is stable (i.e., the distribution of training samples does not change during the training stage). SSGANs can learn high-level representations due to the fact that the self-supervised task (e.g., rotation prediction) requires the discriminator to extract the high-level features. These claims are also validated by the experimental results on representation learning experiments. As shown in Table 2, SSGANs (including SSGAN-LA) outperform BigGAN and DAGAN-MD on Blocks 3-5, showing strong capability of learning high-level representations. As shown in Figure 6 in Appendix C.3, SSGAN-LA achieves consecutive increase in accuracy, showing that SSGAN-LA does not encounter the discriminator forgetting issue. We therefore argue that SSGAN-LA learned stable representations to achieve this. For more clarifications on learning stable representations, please refer to the introduction of SSGAN paper [5].

---

> ### Comment · Reviewer_f1xR · 2021-08-31
> **Thanks for the replies.**
>
> I thank the authors for the replies. I increased my score from 6 to 7. I do not have further questions.

---

### Official Review · Reviewer_NbmS · 2021-07-16

**Rating:** 5
**Confidence:** 4

**Summary:**


The authors argue that GAN discriminators experience catastrophic forgetting due to instability during training (the exact source of the instability the authors refer to is not clear). The authors claim that self-supervised methods, that provide an auxiliary loss (via an auxiliary self-supervised task),for GAN training can help prevent catastrophic forgetting, however they point out two weaknesses with the current methods:
(1) The auxiliary self-supervised losses do not consider the data synthesised by the generator.
(2) They have additional hyper-parameters that need to be fine-tuned to balance losses.

In this paper the authors train the self-supervised classifier on the transformed generate data as well as the transformed real data. Rather than having a separate classifier the discriminator predict the label, this avoids the need for balance hyper-parameters.

**Limitations And Societal Impact:**

The authors do not discuss the limitations of their work.

**Main Review:**

**Originality**

- The authors extend self supervised GANs to learn a join distribution p(real, label) rather than learning each independently.

**Quality**

- It’s good that the authors compare to other data augmentation methods to distinguish the benefits of self-supervised training from those of data augmentation.

- Figure 2: It would be helpful if all figures shared the same y axis this would highlight better that benefits of SSGAN-LA over other the baselines.

- Synthetic dataset: in general (and also according to this paper) when training a generative model the objective is to learn a distribution that matches the training data distribution. Given this, I’m unsure why experiments in section 5.1 are trying to recover the distribution of the augmented data? From my understanding of the paper the goal should be to recover the original training data distribution, the data augmentation used in self-supervised learning should only improve the model’s ability to recover that distribution. Ultimately, these results suggest that the SSGAN-LA model learns to generate samples from outside the original training data distribution? Some clarification on this would be really helpful, thank you.

Further in Section 5.2.1, qualitative results: the authors suggest the fact the DAGAN synthesises transformed data samples is a weakness, adding additional confusion to the motivation behind results in Figure 2.

- The authors compare SSGAN-LA to a large and relevant number of baselines models on three natural image datasets.

- Table 1: The authors suggest SSGAN-LA outperforms all baselines on all tasks, reporting FID and IS in all cases. It’s not clear that SSGAN-LA is statistically significantly better than DAGAN-MD on the STL-10 task, may the authors please clarify? How reproducible are these results? How many runs did you do?

- In some cases the authors have put answers to the Checklist in the checklist rather than in the paper. For example the hardware used for training and the implementation details. These should be in the main paper.

- 5.2.2 Image classification, Table 2: the authors show that the discriminator learns representations useful for classification. The SSGAN-LA appears to show some improvement, however the authors do not provide stdev. so it is hard to tell if these improvements are statistically significant. For example, CIFAR-100 SSGAN-MS performance is very close to SSGAN-LA.

**Clarity**

- Line 23: What do the authors mean by a “non-stationary learning environment”? Is there a more precise definition? Are you referring to the learning dynamics or the distribution of synthesised data?

- Line 31: What do you mean by “stable representations”? Are these representations that help to improve stability during training? Or representations that don’t change significantly between training steps?

- Line 48 - 54: From the introduction it’s not clear what the discriminator will predict. Line 48-49 suggests it predicts only the label, but 52-53 suggests it predicts “real” or “fake”

- Line 100: What is meant by “enforces the generator to produce rotation-detectable image”? Do you mean images, whose orientation can be easily predicted (by a classifier)?

- Figure 1: What is SNGAN?

- Proposition 1: To have notation more consistent with the GAN literature,  it would be good to super-script D with a * e.g. D*(k, 1|y) = … .

- Line 159 - 161: The authors claim “Compared with standard GANs, the transformation prediction especially on real samples of SSGAN LA establishes stationary self-supervised signals to prevent the discriminator from the forgetting problem.” - what is means by stationary self-supervised signals? At this point it is still not clear how training signals from the self-supervised tasks help to prevent catastrophic forgetting?

 - Does Theorem 5 apply to DAGAN-MD,  SSGAN-LA or both?

- On a first parse it was not clear that the role of the  “The importance of Self-supervision” section was to introduce baselines? It may make sense to refer to this section as “Related Work” or “Related Work: The importance of Self-supervision” or to include is in the “Experiments” section.

**Significance**

- The approach presented in the paper is an incremental improvement on existing SS-GAN architectures and the improvements.

- There is no clear theoretical contribution in the main body of the paper.

- The link between the proposed approach and catastrophic forgetting are not made clear in the paper.

- The authors do show some improvements on image synthesis, and show the application of the learned representations to image classification.

- The proposed approach does not require a hyper-parameter to trade off between the self-supervision loss and the discriminator loss.

**Time Spent Reviewing:**

6

---

> ### Author Response · Authors · 2021-08-10
> **Response to Reviewer NbmS**
>
> We would like to thank the reviewer for valuable feedback on this paper and approval of our method and our comprehensive baselines. We clarify the misunderstanding of the reviewer on quality and highlight our theoretical contribution. Hopefully our responses will enable the reviewer to have a more accurate evaluation on this work.
>
> ---
>
> **Quality**
>
> ---
>
> **Q1:** Figure 2: It would be helpful if all figures shared the same y axis.
>
> **R1:** We thank the reviewer for providing constructive suggestion on Figure 2, and will adopt the suggestion in the updated version.
>
> ---
>
> **Q2:** Synthetic dataset: I’m unsure why experiments in section 5.1 are trying to recover the distribution of the augmented data?
>
> **R2:** In fact, the generator is only trying to recover the distribution of original data (shown in blue line). In other words, the blue line in figure 2 represents the distribution of untransformed generated data as we labeled in the top right corner. Other three solid lines represent the distributions of transformed generated data, which is not the originally generated data of methods. We plot these transformed distributions for better visualization. We will delete the mixture line in the updated version.
>
> ---
>
> **Q3:** Further in Section 5.2.1, qualitative results: the authors suggest the fact the DAGAN synthesises transformed data samples is a weakness.
>
> **R3:** The Figure 3 shows that DAGAN directly generates transformed data (outside from the original training data distribution), verifying Theorem 4. And SSGAN-LA avoids generating transformed data.
>
> ---
>
> **Q4:** Table 1: It’s not clear that SSGAN-LA is statistically significantly better than DAGAN-MD on the STL-10 task, may the authors please clarify? How reproducible are these results? How many runs did you do?
>
> **R4:** Due to our limited computational resources, conducting a single trial on STL-10 requires about 3 days. Therefore we decided to run methods on STL-10 for one trial in this paper. We conduct three trials of each method and report the results in the following table, which shows that SSGAN-LA significantly outperforms DAGAN-MD on STL-10.
>
> | Methods  | FID                       | IS                        |
> | -------- | ------------------------- | ------------------------- |
> | DAGAN-MD | $15.92 \pm 0.36$          | $10.31 \pm 0.01$          |
> | SSGAN-LA | $\mathbf{14.65} \pm 0.17$ | $\mathbf{10.46} \pm 0.08$ |
>
> ---
>
> **Q5:** The hardware used for training and the implementation details should be in the main paper.
>
> **R5:** We would like to thank the reviewer for the valuable suggestion and will report the hardware and more details of the implementation in the main paper.
>
> ---
>
> **Q6:** It is hard to tell if these improvements are statistically significant. For example, CIFAR-100 SSGAN-MS performance is very close to SSGAN-LA.
>
> **R6:** We here report the comparison between SSGAN-MS and SSGAN-LA on CIFAR-100. The results are calculated by three trials, which show that SSGAN-LA significantly outperforms SSGAN-MS especially on deep layers (block 2-4).
>
> | Methods  | Block-1                      | Block-2                      | Block-3                      | Block-4                      |
> | -------- | ---------------------------- | ---------------------------- | ---------------------------- | ---------------------------- |
> | SSGAN-MS | $0.3199 \pm 0.0013$          | $0.4542 \pm 0.0069$          | $0.4950 \pm 0.0053$          | $0.5138 \pm 0.0048$          |
> | SSGAN-LA | $\mathbf{0.3201} \pm 0.0019$ | $\mathbf{0.4623} \pm 0.0047$ | $\mathbf{0.5077} \pm 0.0033$ | $\mathbf{0.5375} \pm 0.0049$ |
>
> ---
>
> **Clarity**
>
> ---
>
> **Q7:** Line 23: What do the authors mean by a “non-stationary learning environment”?
>
> **R7:** The non-stationary learning environment means that the training dataset of the discriminator is changing in each step due to the distribution of synthesised data is changing in each step.
>
> ---
>
> **Q8:** Line 31: What do you mean by “stable representations”?
>
> **R8:** Stable representations mean the representations that don't change significantly during training and thus can help to improve the training stability of GANs. We adopt this concept from SSGAN, which said "Arguably, augmenting the discriminator with supervised information encourages it to learn more stable representations which opposes catastrophic forgetting" and "To ensure that the representations learned by the discriminator are more stable and useful, we add an auxiliary, self-supervised loss to the discriminator.". As shown in Figure 6 in Appendix C.3, SSGAN-LA achieves consecutive increase in accuracy, showing that SSGAN-LA does not encounter the discriminator forgetting issue. We therefore argue that SSGAN-LA learns stable representations. We will clarify it and cite SSGAN here in the updated version.
>
> ---
>
> **Q9:** Line 48 - 54: From the introduction it’s not clear what the discriminator will predict. Line 48-49 suggests it predicts only the label, but 52-53 suggests it predicts “real” or “fake”
>
> **R9:** Line 48-49 suggests that the discriminator predicts the augmented label and Line 52-53 does not suggest it predicts only real or fake. Specifically, the discriminator of the proposed SSGAN-LA takes the form of $D(k,i|T_k(x))$ (solving the task of $\mathcal{T}\circ\mathcal{X}\rightarrow ${ $T_1^\text{real},T_2^\text{real},\cdots,T_K^\text{real},T_1^\text{fake},T_2^\text{fake},\cdots,T_K^\text{fake}$ }, $K$ is the number of transformations) that recognizes the performed transformation $t$ while distinguishing real ($i=1$) from fake ($i=0$) simultaneously. In other words, the discriminator predicts both the label and "real" or "fake" at the time. We will polish our description in the updated version for better understanding.
>
> ---
>
> **Q10:** Line 100: What is meant by “enforces the generator to produce rotation-detectable image”?
>
> **R10:** As shown in SSGAN-MS, the images whose orientation can be easily predicted by the classifier only occupy a small region of the real images.
>
> ---
>
> **Q11:** Figure 1: What is SNGAN?
>
> **R11:** It should be GAN, we will fix this typo in the updated version.
>
> ---
>
> **Q12:** Proposition 1: It would be good to super-script D with a * e.g. D*(k, 1|y) = … .
>
> **R12:** We thank the reviewer for pointing out the typo and will fix it in the updated version.
>
> ---
>
> **Q13:** Line 159 - 161: What is means by stationary self-supervised signals?
>
> **R13:** We say the self-supervised signals are stationary because that the data distribution of the self-supervised task on real samples is unchanged during training. Notice that the catastrophic forgetting issue is due to the fact that the distribution of training samples is changed, adding the self-supervised task (on real samples) whose distribution of training samples is unchanged would help to prevent catastrophic forgetting. As shown in Figure 6 in Appendix C.3, SSGAN-LA achieves consecutive increase in accuracy, showing that SSGAN-LA does not encounter the discriminator forgetting issue. However, GAN obtains worse accuracy in the latter training stage, meaning that GAN (without self-supervision) suffered from this issue. For more clarification, please refer to SSGAN.
>
> ---
>
> **Q14:** Does Theorem 5 apply to DAGAN-MD, SSGAN-LA or both?
>
> **R14:** Theorem 5 only applies to DAGAN-MD. We will clarify it in the updated version.
>
> ---
>
> **Q15:** Regarding the title of the “The importance of Self-supervision” section.
>
> **R15:** We would like to thank the reviewer for providing the constructive suggestion. Section 4 aims to discuss the importance of self-supervision of the proposed method. We therefore introduce DAGAN ($D(i|T_k(x))$), which is a simplified version of SSGAN-LA ($D(k,i|T_k(x))$) by ignoring the self-supervision $k$, and DAGAN-MD ($D(i|T_k(x),k)$), which receives self-supervision as input, as ablation models of SSGAN-LA.
>
> ---
>
> **Significance**
>
> ---
>
> **Q16:** The approach presented in the paper is an incremental improvement on existing SS-GAN architectures.
>
> **R16:** Our method unifies the GAN task and the self-supervised task into a joint learning framework, while SSGAN and SSGAN-MS belong to a multi-task learning framework. The joint learning framework is not an incremental improvement on a multi-task learning framework. The proposed method is motivated by our novel understanding that the undesired learning objective in existing self-supervised GANs originates from the generator-agnostic classifier.
>
> ---
>
> **Q17:** There is no clear theoretical contribution in the paper.
>
> **R17:** This paper makes three theoretical contributions.
>
> 1. The most important one is Theorem 3, which shows that the proposed SSGAN-LA can guarantee to learn the real data distribution at optima, perfectly solving the problem of SSGAN and SSGAN-MS.
> 2. The second one is Theorem 2, which corrects the wrong version of SSGAN-MS. In particular, the SSGAN-MS's authors mistakenly regard $\frac{p_d^T(y)}{p_g^T(y)}=\frac{\sum_k p_d(y|T_k)p(T_k)}{\sum_k p_g(y|T_k)p(T_k)}$ as $\frac{p_d(y|T_k)}{p_g(y|T_k)}$ in their proof to derive their wrong result (Theorem 2 in SSGAN-MS). For more details, please compare our proof (in Appendix A.2) with theirs (in Appendix of SSGAN-MS).
> 3. The last one is Theorem 4, which shows that pure data augmentation GANs without utilizing self-supervision could suffer from the convergence point issue, verifying the importance of predicting the self-supervision.
>
> ---
>
> **Q18:** The link between the proposed approach and catastrophic forgetting are not made clear in the paper.
>
> **R18:** Figure 6 in Appendix C.3 shows that the proposed SSGAN-LA does not encounter the discriminator forgetting issue as it achieves consecutive increase in terms of accuracy. We will clarify it in the updated version.

---

> > ### Comment · Reviewer_NbmS · 2021-09-08
> > **Thank you for addressing my comments**
> >
> > The authors have addressed most of my concerns around quality.
> >
> > Thank you for clarifying "non-stationary learning environment" this should be included in a revised version of the paper.
> >
> > Thank you also for clarifying "stable representations", this should also be included in a revised version of the paper. However, it seems that the connection between stability and forgetting is not clear. In the SSGAN paper this connection appears to be a possible explanation, rather than the explanation. Further, it's not clear which "representations" these are? Is it the latent? is it the weights of the model? A more accurate way to determine compare how "stable" representations of different models are is to compare how much they change given some perturbation.
> >
> > In summary, the link between "stable representations" and "catastrophic forgetting" is not clear. Further, there were no clear results in the main body of the paper that demonstrated that the proposed model dealt better with forgetting than previous models despite this being a major motivation for this work.
> >
> > Finally, if the theories are core parts of the authors' contribution then these should be in the main body of the paper. It makes it easier to evaluate the contributions.
> >
> > Overall, I suggest that the authors are more clear about their motivation and their contributions and ensure that their experiments align closely with those contributions.
> >
> > I will increase my score to a 5.

---

> > > ### Author Response · Authors · 2021-09-09
> > > **Thank you and further clarification**
> > >
> > > Dear Reviewer NbmS,
> > >
> > >
> > >
> > > We would like to thank you for providing valuable feedback on this paper. We are encouraged that we have addressed most of your concerns around quality. We hereby respond to your rest concerns by further elaborating our motivation, contributions, and how our experiments verify them.
> > >
> > >
> > >
> > > **Non-stationary learning environment**
> > >
> > > We will add our clarification on the "non-stationary learning environment" as responded in R7 in the revised version.
> > >
> > >
> > >
> > > **Stable representations**
> > >
> > > The "representations" indicate the representation of data learned by the discriminator. Specifically, they can be the output from the hidden layers in the discriminator. And they are related to the weights of the discriminator. The representations are "stable" meaning that the previously learned representations of data would be not affected significantly by the later training samples. This stability is with respect to the training iterations rather than perturbations of data. We will add this clarification in the revised version.
> > >
> > >
> > >
> > > **The link between stable representations and catastrophic forgetting issue**
> > >
> > > The catastrophic forgetting issue means that the discriminator forgets the previously learned knowledge about data with the increase of training iteration. According to the above definition of stable representations that would be not affected significantly by the later training samples, if a discriminator learns stable representations, then it does not suffer from the catastrophic forgetting issue. Figure 6 in Appendix shows that SSGAN-LA does not suffer from the forgetting issue. We will put figure 6 in the main body in the revised version.
> > >
> > >
> > >
> > > **Major motivation of this work**
> > >
> > > The major motivation for this work is to eliminate the undesired goal of existing self-supervised GANs (SSGAN and SSGAN-MS) while improving the training stability (by mitigating the discriminator forgetting issue) of the standard GAN. SSGAN (and SSGAN-MS) mitigated the forgetting issue (and thus can improve the training stability) of the standard GAN by introducing an auxiliary self-supervised task but brought an undesired goal inconsistent with the standard GAN. Our SSGAN-LA inherits the best of both worlds by seamlessly combining the standard GAN task with the self-supervised task. Figure 6 shows that SSGAN-LA does not suffer from the forgetting issue (even better than SSGAN). Theorem 3 shows that SSGAN-LA can guarantee the generator to learn the real data distribution at the optima like standard GAN.
> > >
> > >
> > >
> > > **Theoretical contributions and experimental support**
> > >
> > > We are encouraged that you acknowledge our theoretical contributions made in this paper. Actually, the major theorems are already in the main body of the paper. For example, the most important theorem 3 is located in the Methods Section, which shows that our method solves the problem of existing SSGANs (shown in theorem 1 and theorem 2). Figure 2 in the Experiments Section and Figure 5 in Appendix verify that our SSGAN-LA could faithfully replicate the real data distribution, while SSGAN and SSGAN-MS cannot. These results accord with theorem 1,2,3. In addition, experimental results on real-world datasets also demonstrate the superiority of the proposed SSGAN-LA compared to competing methods.
> > >
> > >
> > >
> > > We thank you again for your time and effort in reviewing our paper. We sincerely hope that our responses would address your concerns on this paper.
> > >
> > >
> > >
> > > Best regards,
> > >
> > > The authors

---

### Decision · Program_Chairs · 2021-09-27

**Decision:**

Accept (Poster)

**Comment:**

This was a borderline paper with a good amount of discussion both between the reviewers and reviewers and authors. I'll try to summarize some of the pros and cons of this work:

* novel yet simple extension to SSGAN. Simplicity is good, but some reviewers have argued that contribution is incremental.
* the theoretical contribution is not in the main body of the paper (this is easily fixed)
* the link with catastrophic forgetting should be made more obvious in the paper itself.
* the results are on small resolution sets only (some reviewers argued that access to resources is not an excuse, but I respectfully disagree).

All in all, I am inclined to accept this work based on the fact that it's a simple, principled, theoretically grounded change that seems to work well in comparable empirical settings on lower-resolution image sets. And I think self-supervised GANs are an interesting practically useful line of research, where novel insights may be useful to many researchers.